# Future evolution and uncertainty of river flow regime change in a deglaciating river basin

Jonathan D Mackay[1,2], Nicholas E Barrand[1], David M Hannah[1], Stefan Krause[1], Christopher R Jackson[2], Jez Everest[3], Guðfinna Aðalgeirsdóttir[4], and Andrew R Black[5]

[1]School of Geography, Earth and Environmental Sciences, University of Birmingham, Edgbaston, Birmingham, B15 2TT, UK
[2]British Geological Survey, Environmental Science Centre, Keyworth, Nottingham, NG12 5GG, UK
[3]British Geological Survey, Lyell Centre, Research Avenue South, Edinburgh, EH14 4AS, UK
[4]Institute of Earth Sciences, University of Iceland, 101 Reykjavík, Iceland
[5]Geography and Environmental Science, University of Dundee, Dundee, DD1 4HN, UK

**Correspondence:** Jonathan D Mackay (joncka@bgs.ac.uk)

**Abstract.** The flow regime of glacier-fed rivers are sensitive to climate change due to strong climate-cryosphere-hydrosphere interactions. Previous modelling studies have projected changes in annual and seasonal flow magnitude, but neglect other changes in river flow regime that also have socio-economic and environmental impacts. This study employs a signature-based analysis of climate change impacts on the river flow regime for the deglaciating Virkisá river basin in southern Iceland. 25 metrics (signatures) are derived from 21st century projections of river flow time-series to evaluate changes in different characteristics (magnitude, timing and variability) of river flow regime over sub-daily to decadal timescales. The projections are produced by a model chain that links numerical models of climate and glacio-hydrology. Five components of the model chain are perturbed to represent their uncertainty including the emission scenario, numerical climate model, downscaling procedure, snow/ice melt model and runoff-routing model. The results show that the magnitude, timing and variability of glacier-fed river flows over a range of timescales will change in response to climate change. For most signatures there is high confidence in the direction of change, but the magnitude is uncertain. A decomposition of the projection uncertainties using analysis of variance (ANOVA) shows that all five perturbed model chain components contribute to projection uncertainty, but their relative contributions vary across the signatures of river flow. For example, the numerical climate model is the dominant source of uncertainty for projections of high-magnitude, quick-release flows, while the runoff-routing model is most important for signatures related to low-magnitude, slow-release flows. The emission scenario dominates mean monthly flow projection uncertainty, but during the transition from the cold to melt season (April and May) the snow/ice melt model contributes up to 23% of projection uncertainty. Signature-based decompositions of projection uncertainty can be used to better design impact studies to provide more robust projections.

## 1 Introduction

Mountain watersheds have been referred to as the world's water towers (Viviroli and Weingartner, 2004; Viviroli et al., 2007), partly because they receive large quantities of precipitation relative to adjacent lowlands, but also because they regulate runoff

through the seasonal accumulation and melt of snow and ice. The presence of snow and ice profoundly affects characteristics of downstream river flow regime including flow magnitude, timing and variability over a range of timescales (Jansson et al., 2003; Mankin et al., 2015). This is partly due to the periodic (diurnal and seasonal) variations and longer-term (decadal) trends in melt water inputs brought about by fluctuations in glaciological mass balance. In addition, the dynamic water storage and realease properties of snow and ice (runoff-routing) control downstream river flow response to runoff over hourly to seasonal timescales (Willis, 2005). As such, glaciated basins exhibit river flow regimes that differ from their non-glaciated equivalents. For example, the so-called 'compensation effect' has been widely observed in the northern hemisphere, whereby partially-glaciated catchments demonstrate reduced intra-annual flow variability (Fountain and Tangborn, 1985; Chen and Ohmura, 1990). Indeed, the compensation of runoff from melt inputs can actually serve to increase mean runoff during anomalously dry heat wave events (Zappa and Kan, 2007).

Mountain glaciers are retreating at unprecedented rates (Zemp et al., 2015) while snow coverage is receding (Vaughan et al., 2013) resulting in observable changes to downstream river flows (Luce and Holden, 2009; Singh et al., 2016; Hernández-Henríquez et al., 2017; Matti et al., 2017). With near-surface air temperature projected to rise over the coming decades (Collins et al., 2013) future changes in river flow regimes in response to cryosphere change could have wide-ranging socio-economic and environmental impacts. Long-term reductions in melt water inputs will disrupt the supply of water available for irrigation (Nolin et al., 2010; McDowell and Hess, 2012; Carey et al., 2014; Baraer et al., 2015). Increased inter-annual and intra-annual flow variability will threaten infrastructure projects such as hydroelectric power stations (Laghari., 2013; Gaudard et al., 2014; Carvajal et al., 2017). The loss of the runoff-regulating effects of snow and ice could result in more frequent short-term very high flows putting downstream populations and infrastructure at risk (Laghari., 2013; Stoffel et al., 2016). Changes in flow magnitude and variability from annual to sub-daily timescales will threaten the sustainability of some of the world's most pristine freshwater ecosystems (Bunn and Arthington, 2002; Naiman et al., 2008; Beamer et al., 2017). Therefore, it is of paramount importance to make reliable projections of changes in downstream river flow regimes from glaciated watersheds so that future impacts can be adapted to and mitigated.

Computational glacio-hydrological models (GHM) driven by numerical climate model projections allow us to assess how future river flow regimes will change in glaciated river basins. Past studies have focussed on projecting changes in decadal, annual and seasonal variations in runoff magnitude. Decadal changes in runoff are inevitable over the coming century (e.g. Bliss et al., 2014; Lutz et al., 2014; Shea and Immerzeel, 2016) where enhanced melt will result in increased river discharge to a point in time termed 'peak water' after which the continued loss of snow and ice will result in an overall decrease in river flow. It has been shown that many basins, particularly those with small glaciers, have already reached peak water and face a future of dwindling water supply (Huss and Hock, 2018). Seasonal flow magnitudes are also projected to change as melt cycles evolve and watersheds shift from glacial-nival to pluvial runoff regimes (Kobierska et al., 2013; Duethmann et al., 2016; Ragettli et al., 2016; Garee et al., 2017).

Some impact studies show robust changes in the magnitude of the highest and lowest river flows including Wijngaard et al. (2017) who projected an increase in the magnitude of the 10% exceedance flow ($Q_{10}$) for river basins across the Hindu-Kush-Himalayan region. Other studies for the Rhine (Bosshard et al., 2013), upper Indus (Lutz et al., 2016) and upper Yellow river

basin (Vetter et al., 2015) show high flow magnitudes will increase. Stewart et al. (2015) projected a decrease in low flow magnitude ($Q_{90}$) for the snow-covered Sierra Nevada and Upper Colorado river basins due to shifts in the snowmelt season and changes in precipitation type from snow to rain. For the Hindu-Kush, Wijngaard et al. (2017) found the opposite impact with an increase in the magnitude of low flow events. The projected trends in $Q_{90}$ for the upper Yellow river basin by (Vetter et al.,

2015) were inconclusive as they showed an even spread of positive and negative trends under the warmest climate scenarios.

Of course, one could go beyond projecting changes in seasonal to decadal mean flow magnitudes and quantiles of the flow duration curve (FDC). A branch of streamflow analysis that has been widely adopted in hydrology is the calculation of river flow 'signatures' which are metrics derived from river discharge time-series that represent different characteristics of river flow over specific timescales. These may include mean flows and FDC quantiles as well as metrics to quantify the variability (e.g.

coefficient of variation), timing (e.g. peak flow month) and flashiness (e.g. autocorrelation) of flows. Signatures have been used in the past to analyse catchment runoff behaviour and similarity (Yadav et al., 2007; Ali et al., 2012). Furthermore, their ability to localise specific aspects of runoff behaviour make them ideal diagnostic evaluation metrics for model hypothesis testing (Euser et al., 2013; Coxon et al., 2014; Hrachowitz et al., 2014) and calibration (Hingray et al., 2010; Shafii and Tolson, 2015; Kelleher et al., 2017; Schaefli, 2016). They also offer an opportunity to evaluate past (Sawicz et al., 2014) and future (Casper

et al., 2012) river flow regime change. For example, Teutschbein et al. (2015) projected changes in 14 different river flow signatures for 14 snow-covered catchments in Sweden and showed daily to annual river flow magnitude, timing and variability were all sensitive to climate change. An analysis like this is yet to be undertaken for any glaciated river basins.

Projections of river flow regime are inherently uncertain due to assumptions made about the formulation, parameterisation and boundary conditions of the underlying GHM (Ragettli et al., 2013; Huss et al., 2014; Jobst et al., 2018) and climate

model, be that a general circulation model (GCM), or combined GCM and regional climate model (GCM-RCM) (Giorgi et al., 2009). Uncertainties may also be introduced by intermediary steps employed to link the two sets of models together such as downscaling (DS). Quantifying the propagation of uncertainties from all sources in the model chain provides a basis for assigning more robust levels of confidence to river flow projections. Additionally, one can assess the relative contributions of model chain components to the total projection uncertainty, providing empirical evidence for future research needs (e.g. Meresa

and Romanowicz, 2017). Ensemble-based experiments have been used in the past to provide this understanding. Here, different components of the model chain are perturbed, typically using a 'one at a time' (OAT) approach where the spread in projections for each perturbed component is evaluated. Ragettli et al. (2013) perturbed three components of a model chain applied to the Hunza River Basin, northern Pakistan including the GCM, statistical DS model and parameterisation of the GHM. They showed that all three sources contributed to annual runoff projection uncertainty, but for the heavily glaciated sub-regions of

the catchment, the GHM parameter uncertainty exceeded the effect of other sources. Huss et al. (2014) investigated uncertainty in seasonal river flow projections over the 21st century for the Findelengletscher catchment, Switzerland by modifying the GCM-RCM, GHM melt model structure and initial ice volume boundary condition. Of these, they found that the GCM-RCM and initial ice volume were most important while the melt model structure was of secondary importance. Jobst et al. (2018) investigated uncertainties in 21st century river flow projections for the Clutha river basin, New Zealand. They evaluated

contributions from emission scenario (ES), GCM-RCM, statistical DS approach and melt model structure. Similarly to Huss et al. (2014), they found that uncertainty in the choice of GCM-RCM dominated total projection uncertainty.

The OAT method provides a first-order approximation of the relative contribution of each component to the total projection uncertainty. However, findings are dependent on how the non-perturbed model components are fixed. Furthermore, this approach cannot resolve interactions between model components which may also contribute to projection uncertainty (Pianosi et al., 2016). The Analysis of Variance (ANOVA) statistical method (von Storch and Zwiers, 1999; Tabachnick and Fidell, 2014) addresses these shortcomings and has been adopted in a number of recent regional and global scale hydrological modelling studies (Bosshard et al., 2013; Addor et al., 2014; Giuntoli et al., 2015; Vetter et al., 2015; Samaniego et al., 2017; Vetter et al., 2017; Yuan et al., 2017) to compare uncertainties stemming from ES, climate model, hydrological model structure and DS approach. While uncertainties associated with future climate tend to dominate projections of river flow, glacier-fed river flow projections have shown to be highly sensitive to hydrological model structure (Addor et al., 2014; Giuntoli et al., 2015), particularly in relation to high flows (Vetter et al., 2017). Furthermore, the contribution of projection uncertainty from interactions between model chain components can exceed individual components (Bosshard et al., 2013; Addor et al., 2014; Vetter et al., 2015). Several issues not considered in these studies, however, are yet to be addressed. Firstly, none have investigated a full range of characteristic changes in river flow regime covering decadal to sub-daily timescales. Second, all have incorporated hydrological model uncertainty using multiple model codes, each with their own unique set of process representations, resolution, timestep and climate interpolation strategies making it difficult to determine which model components contribute most to projection uncertainty. Finally, none included a fully integrated mass-conserving, dynamic glacier evolution model component and therefore could not fully account for atmosphere-cryosphere-hydrosphere feedbacks.

This study uses a GCM-RCM-DS-GHM model chain to simulate the impact of 21st century climate change on downstream river flow regime in the deglaciating Virkisá river basin in southern Iceland. Five components of the model chain are perturbed to represent uncertainty of ES, GCM-RCM, statistical DS parameterisation and structure-parameterisation of two primary controls on river flow regime in the GHM: melt and runoff-routing processes. The study has two principal aims: i) to determine how climate change and consequent cryospheric change will impact on downstream river flow regime over the 21st century; and ii) to quantify the relative influence of the five model chain components to projection uncertainty across the different characteristics of river flow regime. This study addresses each of the aforementioned gaps in previous work. Firstly, changes in river flow regime are assessed quantitatively using 25 river discharge signatures which define different characteristics of river flow regime over a range of timescales. Second, a single, consistent, GHM code is used that can incorporate different model structures and parameterisations of melt and runoff-routing processes allowing for uncertainty stemming from these to be localised using ANOVA. Finally, a fully integrated mass-conserving, dynamic glacier evolution routine is included in the GHM code.

## 2 Methodology

### 2.1 Study site

The Virkisá river basin covers an area of 22 km$^2$ on the western side of the ice-capped Öræfajökull stratovolcano in south-east Iceland (Figure 1) and forms a primary drainage channel for accumulating ice at the mountain summit ($\sim$ 2000 m asl). The glacier flows in a south-westerly direction (average ice surface slope = 0.25) along two distinct glacier arms, Virkisjökull and Falljökull, (hereafter referred to as Virkisjökull) around a bedrock ridge before meeting again at the terminus ($\sim$ 150 m asl). Virkisjökull currently covers $\sim$ 60% of the river basin area, but has been in a phase of retreat since 1990. Between 1988 and 2011 Virkisjökull lost $\sim$ 0.3 km$^3$ of ice and retreated $\sim$ 0.5 km. A small proglacial lake at the terminus forms the headwater of the Virkisá River. The Virkisá flows through an 800 m bedrock-controlled section flanked on either side by push moraines and then over the Skeiðarársandur floodplain typically comprising unconsolidated glacial outwash sediment. The steep-sided valley walls and glacial activity only allow for sporadic development of thin soils with limited vegetation including mosses, grass and shrubs.

The local climate is characterised by cool summers ($\sim$10 °C on average at AWS1) and mild winters ($\sim$1 °C on average at AWS1) with an average temperature lapse rate of -0.44 °C 100 m$^{-1}$ (Flett, 2016). There is a significant lateral precipitation gradient due to the prevailing north-easterly winds and orographic effects with more than five-times the annual precipitation falling at the Öræfajökull summit ($\sim$8000 mm yr$^{-1}$) compared to lower down at the catchment outlet to the west ($\sim$1500 mm yr$^{-1}$) (Nawri et al., 2017).

### 2.2 Climate data

#### 2.2.1 Historical climate

Historical climate data were available from 1981 to 2016 inclusive. A detailed description of these are provided by Mackay et al. (2018). For brevity, only a summary of these data are provided here. The historical climate data include continuous hourly near-surface air temperature measurements from two automatic weather stations (AWS) in the catchment (Fig. 1c) which were installed in 2009 (AWS1) and 2011 (AWS4). Temperature data from the nearby Icelandic Meteorological Office Fagurhólsmýri weather station (12 km south of the study site) were used to extend the AWS1 time-series back to 1981 using a linear regression model (R$^2$=0.92) to bias-correct against the AWS data. A seasonally variable hourly lapse rate calculated between AWS1 (156 m asl) and AWS4 (805 m asl) was used to extrapolate near-surface air temperature across the study region and an on-ice temperature correction function (Shea and Moore, 2010) was employed to account for katabatic cooling of air in the glacier valleys. Continuous hourly incident solar radiation data was also available from AWS1. A random resampling strategy that accounted for the dependence between intra-day solar radiation and temperature variability was employed to generate a continuous time-series back to 1981. For precipitation, the 2.5 km gridded hourly total precipitation data produced as part of the ICRA atmospheric reanalysis project were used, which are currently considered the most accurate gridded

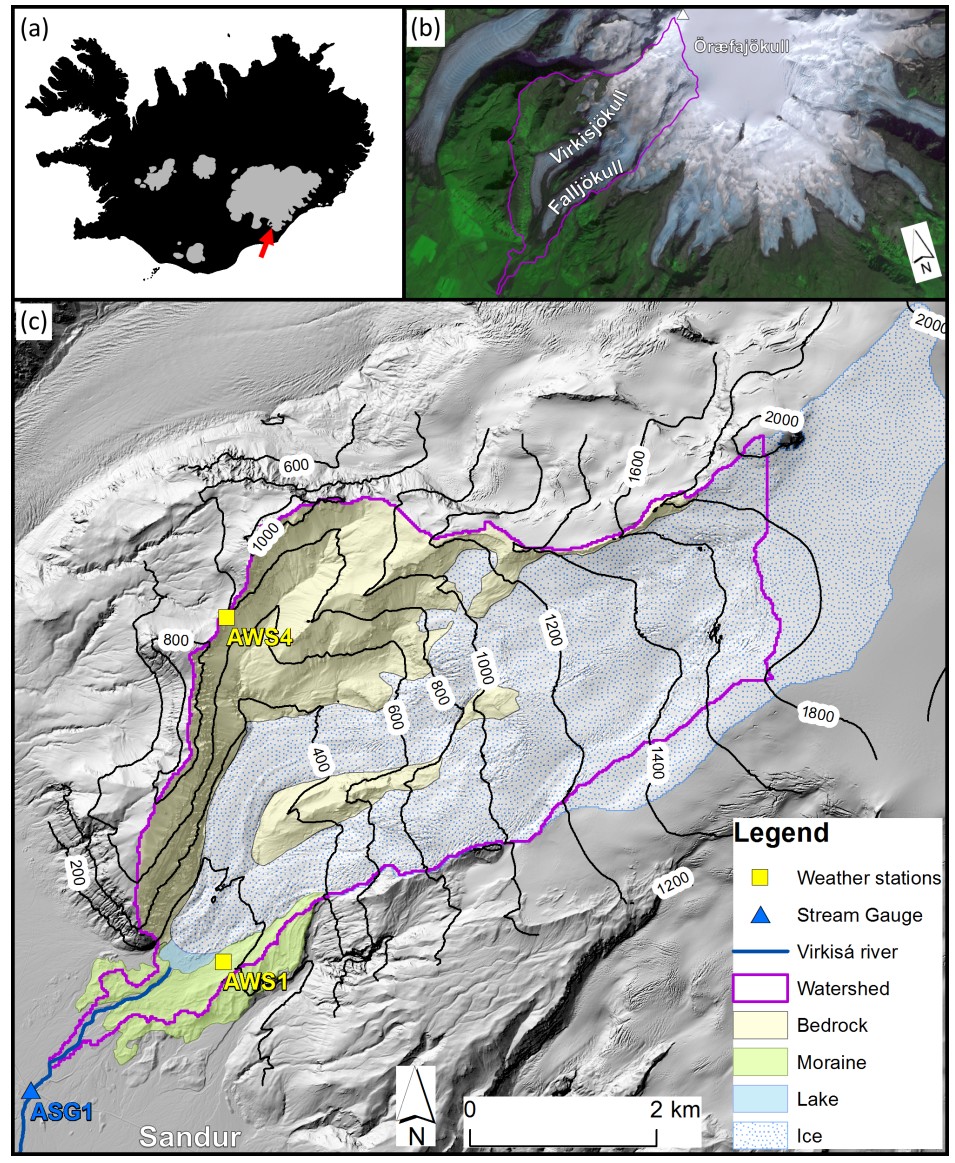

**Figure 1.** Location of Virkisá river basin in Iceland with glaciated areas highlighted in grey (a); on Öræfajökull (b); and detailed topographical map of basin including major land surface types and meteorological and stream gauging stations (c).

precipitation product over Iceland (Nawri et al., 2017). These were bias-corrected against hourly precipitation measurements from AWS1 using equidistant quantile mapping (Li et al., 2010).

### 2.2.2 Regional climate projections

Future climate time-series until 2100 were constructed using regional climate projections from the Coordinated Regional
Climate Downscaling Experiment (CORDEX) (Giorgi et al., 2009). These are based on an ensemble of RCMs driven by GCM projections from the Coupled Model Intercomparison Project (CMIP5) (Taylor et al., 2012). Iceland is covered by the EURO-CORDEX and ARCTIC-CORDEX regional model domains. Following the review by Gosseling (2017), the EURO-CORDEX data were used as these include projections at a higher $0.11°$ spatial resolution and a larger ensemble of GCM-RCM combinations allowing better exploration of climate model uncertainty.

The $0.11°$ EURO-CORDEX simulations span the years 1950-2100 with simulations up to 2005 constituting the 'recent past' where influences such as atmospheric composition, solar forcing and emissions are imposed based on observations. From 2006, three future ESs or Representative Concentration Pathways (RCPs) were imposed on the models including RCP2.6, RCP4.5 and RCP8.5 which represent an additional radiative forcing by 2100 relative to pre-industrial values of +2.6, +4.5 and +8.5 W m$^{-2}$ respectively. All simulations are available at 3-hourly to 3-monthly resolution, however the 3-hourly simulations
were only produced using 4 GCM-RCMs while daily to seasonal simulations were produced using 15. Given the intent of this study to analyse projection uncertainty, it was decided that the daily data were most suitable. RCP2.6 was omitted as only 8 of 15 GCM-RCMs within the CORDEX archive used this ES. Furthermore, the probability of achieving the RCP2.6 targets is increasingly unlikely (Sanford et al., 2014; Fyke and Matthews, 2015) and arguably completely infeasible (Mora et al., 2013) given the current global emission trajectory.

One of the 15 GCM-RCMs (GCM:CNRM-CM5, RCM:CNRM-ALADIN53) was removed from the ensemble given that it showed an extreme negative winter temperature bias and a consistently low skill when compared to daily observed climate data (see Appendix A). Figure 2 shows the seasonal bias of each of the 14 remaining GCM-RCMs when compared to observations between 1981 and 2005. For temperature, the coldest days ($T_1$) typically show a negative bias, particularly in winter, spring and autumn. Biases for $T_{99}$ are generally positive, but smaller in magnitude. The average absolute bias in mean seasonal
temperature ($T_{mean}$) is 1.4 °C, but the majority of GCM-RCMs show absolute biases <1.2 °C. Biases in seasonal incident solar radiation projections are almost exclusively positive with the largest biases associated with $SW_{mean}$ and $SW_{99}$, particularly in spring and summer where they can exceed 100 W m$^{-2}$. Total precipitation biases are typically largest in winter and autumn where proportionally, biases in $P_{mean}$ can exceed the magnitude of the observations (see SON for [EC-EARTH]-[HIRHAM5]). The largest biases however are seen in extremes ($P_{99}$) which range from -86.9 to 77.5 mm d$^{-1}$. While positive and negative
precipitation biases are present throughout the ensemble, the sensitivity of precipitation simulations to the RCM is clear. For example, the CCLM4-8-17 RCM has a systematic negative bias and the HIRHAM5 RCM has a systematic positive bias.

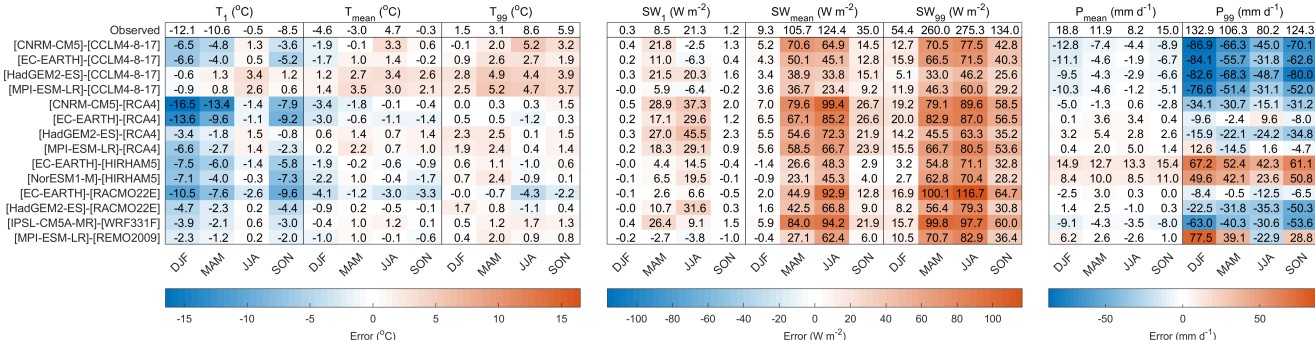

**Figure 2.** Comparison of seasonal catchment-average observed and simulated near surface air temperature (T), incident solar radiation (SW) and total precipitation (P) between 1981 and 2005 for the 14 [GCM]-[RCM] used in this study. The top row shows the observed value and all subsequent rows indicate the GCM-RCM biases. The 1st percentile, mean and 99th are denoted by the subscripts 1, mean and 99 respectively. All statistics are calculated for the recent past (1981-2005) for winter (DJF), spring (MAM), summer (JJA) and autumn (SON).

### 2.2.3 Downscaling regional climate projections

The statistical delta-change downscaling approach was employed which has been widely applied in hydrological impact studies (Farinotti et al., 2012; Immerzeel et al., 2013; Kobierska et al., 2013; Huss et al., 2014; Lutz et al., 2016). While most studies have used monthly mean delta-change values to capture seasonal shifts in climate, several recent investigations have used advanced quantile-based approaches which account for changes in higher-order statistical properties of future climate by evaluating shifts in the ECDFs of climate variables. Including these higher-order changes has shown to be important for evaluating shifts in extreme high flows and sub-seasonal metrics of river flow projections (Jakob Themeßl et al., 2011; Immerzeel et al., 2013; Lutz et al., 2016). In addition, shifts in the day-to-day variability of temperature impact projections of glacier retreat as these variations control the periodic rising of temperature above the melting point (Beer et al., 2018). Accordingly, the advanced delta-change approach was adopted in this study. The approach is summarised in five steps which were applied to each combination of GCM-RCM, climate variable and ES separately:

1. The climate variable time-series was divided into four 25-year long periods including the recent past (1981 - 2005) and early (2006 - 2030), mid (2041-2065) and late (2076 - 2100) 21st century.

2. For each of the four periods, all daily data points were further divided into 12 sub-samples representing each month of the year. An ECDF was constructed for each month of each period.

3. For each month of each future period, ten deltas were calculated by taking the mean difference between the recent past and future ECDF for each 10% section (see grey bars in Fig. 3a for example).

4. Given the need for transient climate time-series to simulate glacier evolution over the 21st century, a daily delta time-series from 2006 to 2100 was constructed for each ECDF section of each month by linearly interpolating between the

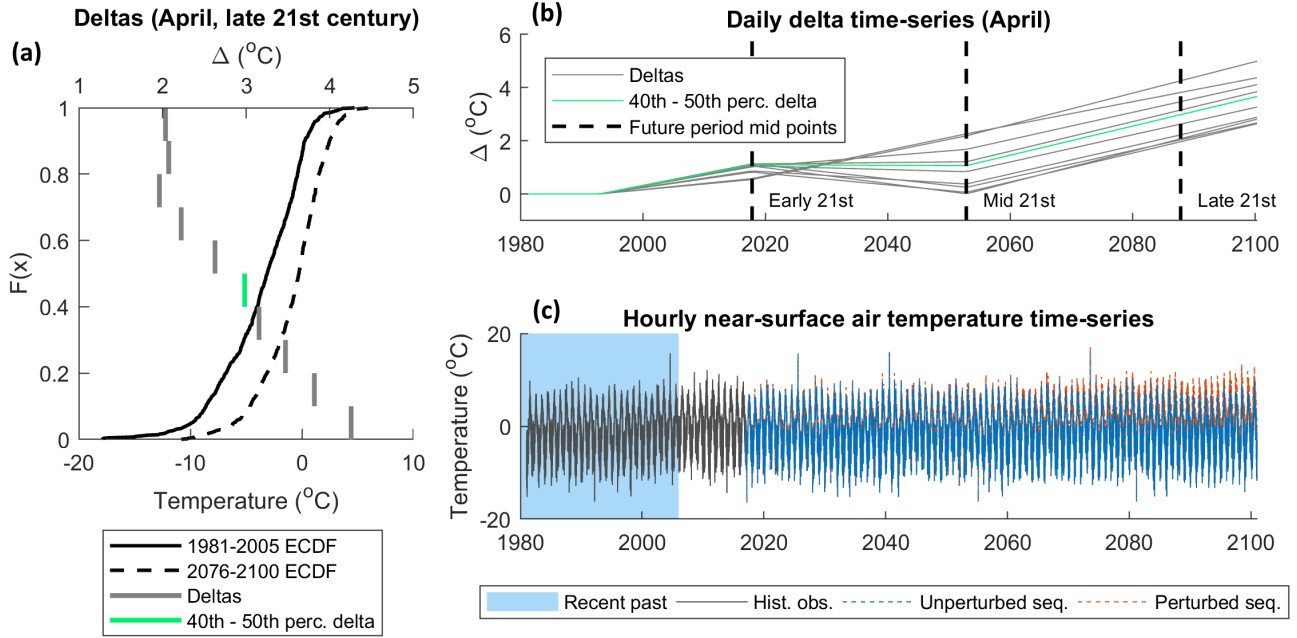

**Figure 3.** Example of advanced delta-change approach when applied to near surface air temperature data based on the RCP8.5 projections using the CNRM-CM5 GCM and CCLM4-8-17 RCM. Deltas (grey bars) derived from ECDFs (black curves) for April in late 21st century (a); Daily delta time-series for each section of the April ECDFs (green line represents 40th - 50th percentile section) (b); Initial and perturbed future temperature time-series when deltas for all months and ECDF sections are applied (c).

calculated deltas of each future period (e.g. as implemented by Farinotti et al., 2012), using the midpoints of the future periods as interpolation points (Figure 3b).

5. The hourly historic observation data for the recent past were randomly sampled (with replacement) on a year-by-year basis to generate an initial unperturbed future climate variable time-series (blue dash, Fig. 3c). The daily deltas were applied to this time-series for each month and ECDF section separately to generate a future perturbed climate time-series at an hourly resolution (orange dash Fig. 3c). It was noted upon visual inspection, that the inter-annual variability of the future climate time-series was very sensitive to the random sampling of the historic climate data. Accordingly, uncertainty associated with this aspect of the DS parameterisation was considered by using ten different random historic climate samples.

For temperature, catchment average daily deltas were applied evenly across the catchment and each daily period of the unperturbed time-series. Accordingly, diurnal temperature variability and lapse rates were assumed not to change in the future. For incident solar radiation and precipitation, proportional deltas were used to prevent negative values and preserve the sub-daily proportional distribution of these variables in space and time. A total of $2\,ES \times 14\,GCM\text{-}RCM \times 10\,DS$ parameterisations $= 280$ future climate time-series were generated for this study.

## 2.3 Glacio-hydrological model

The distributed GHM code implemented by Mackay et al. (2018) was used in this study because it includes a dynamic, mass-conserving glacier evolution component and also allows the user to utilise different model structures of melt and runoff-routing processes. The GHM resolves glacio-hydrological processes over a regular 2D Cartesian grid of 50 m cells driven by hourly climate data including precipitation, temperature and incident solar radiation. Empirical 'index-based' equations simulate the melt of snow and ice. Snow redistribution by drift and avalanches is calculated using the curvature and slope of the surface (Huss et al., 2008) while the mass-conserving $\Delta$h parametrisation of glacier retreat (Huss et al., 2010) resolves changes in the glacier geometry. A soil infiltration and evapotranspiration model (Griffiths et al., 2008) based on the well-established FAO56 soil moisture accounting procedure (Allen et al., 1998) solves the water balance for model nodes with no ice or snow coverage. This model has been applied extensively (Mackay et al., 2014, 2015; Jackson et al., 2016; Mansour et al., 2018) and has shown to compare favourably to physically-based models at the field scale where interception losses are small (Sorensen et al., 2014). Excess soil moisture, rainfall and melt are routed to the catchment outlet via a semi-distributed network of linear-reservoir cascades (Ponce, 1989) which represent the average water storage and release characteristics of the major hydrological pathways in the watershed (see Fig. 2 in Mackay et al., 2018).

### 2.3.1 Modification to $\Delta$h parameterisation of glacier retreat

Under periods of sustained positive mass balance, simulations from the $\Delta$h glacier evolution model may result in an unrealistic build up of ice at the glacier tongue without any simulated areal advance. Given the potential for periods of glacier advance under a changing climate, such behaviour is likely to result in significant projection biases. Recently, Seibert et al. (2018) presented an implementation of the original $\Delta$h parameterisation that provides more realistic simulations of glacier advance. They propose running the $\Delta$h parametrisation a priori outside of the GHM. A small negative mass balance is used to force the $\Delta$h model from an initial glacier profile (ideally its maximum observed extent) until the glacier has disappeared completely. At each step, the glacier mass and geometry are stored in the form of a lookup table. On running the GHM, the retreat/advance of the glacier is derived from the lookup table as a function of the simulated glacier mass. One important drawback of using this static lookup table is that it modifies the behaviour of the $\Delta$h formulation during periods of retreat. More specifically, this approach neglects the transient annual sequencing of glacier mass balance which influences simulated glacier geometry due to the non-linear structure of the $\Delta$h polynomial that defines the relationship between mass balance and glacier geometry. Accordingly, a modified implementation of the Seibert et al. (2018) approach was used in this study which behaves like the original $\Delta$h formulation during periods of glacier retreat and allows for the simulation of glacier advance while accounting for mass balance sequencing effects on the model behaviour. For periods of negative glacier mass balance the original $\Delta$h formulation was used. The GHM was then modified so that for each simulation year, the simulated glacier mass and geometry were stored in memory. If a positive glacier mass balance ($\Delta M$) was simulated, the GHM would log the current glacier mass ($M_{current}$) and then look for the most recent historical simulated glacier mass ($M_{hist}$) that exceeded $M_{current} + \Delta M$. The $\Delta$h model was then run with a negative mass balance ($\Delta M^*$) so that $M_{hist} + \Delta M^* = M_{current} + \Delta M$.

### 2.3.2 Melt and runoff-routing model structures

The selection of melt and runoff-routing model structures was based on the findings of Mackay et al. (2018). They applied nine different combinations of three melt model structures and three runoff-routing model structures of varied complexity in the GHM and evaluated their ability to capture a range of river discharge signatures derived from observation time-series at automatic stream gauge 1 (ASG1 in Fig. 1c) which has been operational since 2012. They also used observation data of ice melt and snow coverage to derive signatures that described different aspects of these data (see Appendix B). They showed that while introducing model complexity did improve simulations when evaluated against specific signatures, it did not necessarily result in better consistency across all signatures, emphasising model selection uncertainty. The most complex runoff-routing structure, however, was consistently the least efficient when compared to the two simpler alternatives, particularly in relation to capturing signatures representing high river flow events. As such, this model structure was discarded, and only the remaining six combinations of melt and runoff-routing models structures were used in this study. These included three melt model structures: i) the classic temperature index model ($TIM_1$) where melt increases linearly with near surface air temperature above a critical threshold (e.g. Braithwaite, 1995); ii) the enhanced temperature-index model ($TIM_2$) proposed by Hock (1999) which accounts for topographic effects on incident solar radiation including surface slope, aspect and shading from the surrounding landform; and iii) the enhanced temperature-index model ($TIM_3$) proposed by Pellicciotti et al. (2005) which accounts for topographic effects and also includes a dynamic snow-albedo parameterisation (Brock et al., 2000) which accounts for the drop in snow albedo as it ages. Each melt model structure was combined with the two runoff-routing structures: i) a single linear reservoir cascade ($ROR_1$) which routes runoff from all sources (ice melt, snowmelt, rainfall and excess soil water) simultaneously; and ii) two linear reservoir cascades in parallel ($ROR_2$) where the first represents the slow percolation of water through the snow and firn and the second represents faster flow of water through and over bare ice and overland. The simplest $ROR_1$ structure assumes all catchment water stores delay and diffuse downstream river response to runoff in the same way, effectively fixing the run-off routing behaviour of the catchment over time. The more complex $ROR_2$ structure accounts for temporal variations in the drainage efficiency of the catchment according to changes in snow and ice coverage.

### 2.4 Signatures of river flow regime

Table 1 lists the 25 signatures of river discharge used to evaluate future changes in river flow regime. The majority of signatures were selected from past studies (Yadav et al., 2007; Yilmaz et al., 2008; Shafii and Tolson, 2015; Schaefli, 2016) and were chosen to reflect the types of changes that one might expect to see in snow and ice covered catchments. They also broadly follow those used in the model assessment study of Mackay et al. (2018). The signatures are grouped into seven different attributes and further categorised by the characteristic(s) of flow regime that they evaluate and their temporal scale. At the decadal timescale, two signatures were selected. These include the 'peak water', which defines the timing (by year) of maximum flow, as well as the inter-annual flow range which characterises long term flow variability. Changes in mean annual river flow were also evaluated, while mean monthly flows were used to evaluate changes to the seasonal timing and magnitude of river flow. The range in mean monthly flows was also chosen to evaluate intra-annual flow variability. In addition, eight signatures were

**Table 1.** Summary of 25 river discharge signatures used to evaluate future changes in river flow regime. Those with available limits of acceptability were also used as part of the GHM calibration and evaluation procedure.

| Attribute | Signature | Limits of acceptability | | Regime characteristic | Temporal scale |
|---|---|---|---|---|---|
| | | Calibration (2013-2014) | Evaluation (2015-2016) | | |
| Inter-annual flow | Peak water (PW) | - | - | Timing and magnitude | Decadal |
| | Inter-annual flow range ($R_{ANN}$) | - | - | Variability | Decadal |
| Annual river flows | Mean annual river flow ($\bar{Q}$) | - | - | Magnitude | Annual |
| Monthly river flows | Mean January river flow ($\bar{Q}_{JAN}$) | 1.16 – 1.86 $m^3\ s^{-1}$ | - | Timing and magnitude | Monthly |
| | Mean February river flow ($\bar{Q}_{FEB}$) | 1.69 – 2.92 $m^3\ s^{-1}$ | - | Timing and magnitude | Monthly |
| | Mean March river flow ($\bar{Q}_{MAR}$) | 0.85 – 1.58 $m^3\ s^{-1}$ | 1.22 - 2.34 $m^3\ s^{-1}$ | Timing and magnitude | Monthly |
| | Mean April river flow ($\bar{Q}_{APR}$) | 0.73 – 1.48 $m^3\ s^{-1}$ | 1.03 - 2.10 $m^3\ s^{-1}$ | Timing and magnitude | Monthly |
| | Mean May river flow ($\bar{Q}_{MAY}$) | 1.50 - 2.16 $m^3\ s^{-1}$ | 1.64 - 3.00 $m^3\ s^{-1}$ | Timing and magnitude | Monthly |
| | Mean June river flow ($\bar{Q}_{JUN}$) | 4.12 – 6.23 $m^3\ s^{-1}$ | 4.88 - 9.39 $m^3\ s^{-1}$ | Timing and magnitude | Monthly |
| | Mean July river flow ($\bar{Q}_{JUL}$) | 6.33 – 10.3 $m^3\ s^{-1}$ | 4.96 - 9.38 $m^3\ s^{-1}$ | Timing and magnitude | Monthly |
| | Mean August river flow ($\bar{Q}_{AUG}$) | 5.72 – 9.15 $m^3\ s^{-1}$ | 6.80 - 14.39 $m^3\ s^{-1}$ | Timing and magnitude | Monthly |
| | Mean September river flow ($\bar{Q}_{SEP}$) | 4.55 – 7.38 $m^3\ s^{-1}$ | 6.61 - 14.21 $m^3\ s^{-1}$ | Timing and magnitude | Monthly |
| | Mean October river flow ($\bar{Q}_{OCT}$) | 3.88 – 7.02 $m^3\ s^{-1}$ | 6.94 - 16.33 $m^3\ s^{-1}$ | Timing and magnitude | Monthly |
| | Mean November river flow ($\bar{Q}_{NOV}$) | 3.90 – 7.40 $m^3\ s^{-1}$ | 3.17 - 5.76 $m^3\ s^{-1}$ | Timing and magnitude | Monthly |
| | Mean December river flow ($\bar{Q}_{DEC}$) | - | - | Timing and magnitude | Monthly |
| | Mean monthly flow range ($R_{mnth}$) | - | - | Variability | Seasonal |
| Slow release low flows | 95% exceedance flow ($Q_{95}$) | 0.27 - 1.10 $m^3\ s^{-1}$ | 0.66 - 1.75 $m^3\ s^{-1}$ | Magnitude | Monthly to seasonal |
| | 99% exceedance flow ($Q_{99}$) | 0.12 - 0.88 $m^3\ s^{-1}$ | 0.46 - 1.56 $m^3\ s^{-1}$ | Magnitude | Monthly to seasonal |
| | Low flow standard deviation ($\sigma_{99-95}$) | 0.03 - 0.10 $m^3\ s^{-1}$ | 0.02 - 0.09 $m^3\ s^{-1}$ | Variability | Monthly to seasonal |
| Moderate flows | 50% exceedance flow ($Q_{50}$) | 2.38 - 3.70 $m^3\ s^{-1}$ | 3.10 - 5.79 $m^3\ s^{-1}$ | Magnitude | Daily to monthly |
| | Moderate flow standard deviation ($\sigma_{52-48}$) | 0.07 - 0.15 $m^3\ s^{-1}$ | 0.08 - 0.18 $m^3\ s^{-1}$ | Variability | Daily to monthly |
| Quick release high flows | 1% exceedance flow ($Q_{01}$) | 17.71 - 40.31 $m^3\ s^{-1}$ | 21.90 - 61.57 $m^3\ s^{-1}$ | Magnitude | Hourly to daily |
| | 5% exceedance flow ($Q_{05}$) | 9.43 - 15.76 $m^3\ s^{-1}$ | 11.71 - 27.37 $m^3\ s^{-1}$ | Magnitude | Hourly to daily |
| | High flow standard deviation ($\sigma_{05-01}$) | 2.08 - 5.68 $m^3\ s^{-1}$ | 2.60 - 8.10 $m^3\ s^{-1}$ | Variability | Hourly to daily |
| Flashiness | Integral scale ($\tau$) | 25 – 44 hr | 0 - 54 hr | Timing | Hourly to daily |

selected which broadly describe the magnitude and variability of slow release low flows (99-95% exceedance flows), moderate flows (52-48% exceedance) and quick release high flows (5-1% exceedance). For these, the quantiles of the FDC were used to assess changes in the magnitude of these flow types. The standard deviation was also used to define flow variability of each flow type. Finally, the integral scale, which measures the lag time at which the autocorrelation function of the river flow time-series falls below $\frac{1}{e}$ was utilised as an indicator of the response time of the catchment to runoff events (flashiness).

## 2.5 GHM calibration

Given the focus on projecting changes in river discharge signatures, these were explicitly included in the GHM calibration procedure as this gives better signature simulations than using traditional global objective functions (Kiesel et al., 2017; Pool et al., 2017). Calibrating against river flow data alone can lead to unrealistic snow and glacier melt rates, inhibiting model consistency and increasing projection uncertainties (Konz and Seibert, 2010; Finger et al., 2011; Schaefli and Huss, 2011; Hanzer et al., 2016). Accordingly, a novel signature-based calibration of the GHM was undertaken by evaluating the GHM against 20 of the river discharge signatures in Table 1 for which observation data exists calculated from hourly river discharge

measurements (Macdonald et al., 2016) at the automatic stream gauge (ASG1 in Fig. 1) in combination with 12 signatures of ice melt and snow coverage (Appendix B).

For each signature, model simulations were compared to observations using a continuous acceptability score that is analogous to those used in other signature-based hydrological studies (Coxon et al., 2014; Shafii and Tolson, 2015). This objective function explicitly accounts for uncertainty in the observation signatures, hereafter termed 'limits of acceptability' (LOA), so that decisions about model appropriateness can be made within the uncertainties of observation data. In this study the 95% confidence bounds were used to define the LOA for the river discharge signatures (Table 1) and the ice melt and snow coverage signatures (Table B1). Details of how these were derived can be found in the study of Mackay et al. (2018). The acceptability for signature $j$ is defined as:

$$
s_j = \begin{cases} 0 & low_j \leq sim_j \leq upp_j \\ \frac{sim_j - upp_j}{upp_j - obs_j} & sim_j > upp_j \\ \frac{sim_j - low_j}{obs_j - low_j} & sim_j < low_j \end{cases} \tag{1}
$$

where $obs_j$ and $sim_j$ are the observed and simulated values and $upp_j$ and $low_j$ are the upper and lower LOA. A score of zero indicates that the model captures the signature within the LOA. A non-zero score is given for any simulation that falls outside of the LOA with a sign that indicates the direction of bias and a magnitude that indicates the model's performance relative to the LOA. A score of -3 would indicate that the model underestimates the signature by three times the observation uncertainty. This score therefore does not penalise a model if it falls within the observation uncertainty of a signature. It is also tolerant of projections that fall outside of the LOA where observation uncertainty is high; a desirable attribute given the range of signatures the GHM was evaluated against.

The aim of the calibration was to extract an ensemble of GHM compositions (TIM and ROR structure-parameter combinations) that were most acceptable across the river discharge signatures whilst broadly reproducing the snow coverage and ice melt signatures. This was achieved using a two-stage Monte-Carlo procedure which was devised so that the resultant GHM ensemble reflected the uncertainty in model selection given the known inconsistencies of the GHM across the signatures.

### 2.5.1 Stage 1: TIM calibration

The first stage aimed to extract the optimal TIM compositions (structure-parameter combinations) by calibrating them against the 12 snow coverage and ice melt signatures. Here, for each of the three TIM structures, 5000 TIM parameter sets were drawn from pre-defined uniform distributions (Table C1) using the quasi-random Sobol sampling strategy (Brately and Fox, 1988) to sample the parameter space as efficiently as possible. For each parameter set, the GHM was spun-up for three years from 1985 to 1988 with a static ice geometry fixed to a 1988 ice DEM (Magnússon et al., 2016). The GHM was then run from 1988 to the end of 2016 with a freely evolving glacier geometry.

Given the high degree of glaciation in the study catchment, and its recent rapid retreat, an initial emphasis of the calibration was put on the model's ability to capture the long term glacier volume change signature. Accordingly, only those TIM compositions that captured this signature within the LOA were considered and the rest were discarded. These remaining compositions

were then further refined by evaluating them against the remaining 11 snow and ice signatures. First, the TIM compositions were sorted by structure ($TIM_1$, $TIM_2$, $TIM_3$). Then, for a given TIM structure, the following steps were applied:

1. Find the TIM parameter set(s) that capture the signature within the LOA and discard the rest. If more than one parameter set captures the current signature, go to step 2. If none capture the current signature, discard none and go to step 2.

2. Of the remaining models, find that which best captures the 10 remaining snow and ice signatures overall according to the weighted mean scores obtained in Eq. 1. The weights were applied to ensure that equal preference was given to ice melt and snow coverage signatures.

24 unique TIM compositions were obtained from this calibration stage made up of eight unique parameterisations of each of the three TIM structures. In some cases the same composition was selected more than once which was accounted for by weighting the simulations in the results presented throughout this study.

### 2.5.2 Stage 2: ROR calibration

The second calibration stage aimed to extract the optimal ROR compositions when used in combination with the 24 pre-selected TIM compositions by calibrating them against 20 of the river discharge signatures obtained from observations of river discharge for the years 2013 and 2014 (see signatures with calibration LOA in Table 1). Note, the inter-annual flow signatures and the mean December river flow signatures could not be calculated as there was insufficient observation data. Furthermore, the mean annual river flow and mean monthly flow range were not included as this information was already accounted for in the mean monthly flow signatures. Here, 5000 random ROR parameter sets were drawn for each ROR structure. Each was used in combination with the pre-selected TIM compositions in the GHM. Then, the two steps outlined in calibration stage 1 were applied using the 20 calibration river discharge signatures with two notable differences. Firstly, for each ROR structure and each river discharge signature, rather than selecting a unique ROR parameter set for each of the 24 TIM compositions, a single parameter set was selected based on its mean performance across the 24 TIM compositions. This was done to satisfy the ANOVA requirements so that the TIM and ROR composition uncertainty could be analysed separately. Furthermore, for step 2, the signatures were weighted so that each of the attributes in Table 1 were weighted equally. In total, 14 unique ROR compositions were selected made up of seven unique parameterisations of the $ROR_1$ and $ROR_2$ structures, giving a total of $24 \times 14 = 336$ unique GHM compositions.

### 2.6 ANOVA uncertainty analysis

For the 21st century runs, all 336 GHM compositions were run to the end of 2016 using the historic observed climate to capture the evolving ice geometry as accurately as possible. From 2017 to 2100, the 280 downscaled future climate time-series were used to drive the GHM compositions resulting in 94080 individual model runs. For each model run, projections of watershed snow and ice coverage and the 25 river discharge signatures were extracted for six 21st century 25-year time-slices centred on the 2030s (2023-2047), 2040s (2033-2057), 2050s (2043-2067), 2060s (2053-2077), 2070s (2063-2087) and 2080s (2073-2097). Future changes in these were then calculated relative to a reference 25-year period (1991-2015). This reference period

was chosen because ice-coverage data (used to initialise the GHM) were only available from 1988 and historic climate data were available up to the end of 2016. ANOVA was used to quantify the effect size of the five components of the model chain, hereafter termed *factors*, on each signature for each 21st century time-slice. Note, the peak water (PW) signature can only be calculated taking into account the full projection time-series and, as such, it was not possible to apply ANOVA to each time-slice for this signature. The five factors include the $2 \times$future ES, $14 \times$GCM-RCM combinations, $10 \times$DS parameterisations, $24 \times$TIM compositions and $14 \times$ROR compositions. ANOVA offers an intuitive approach to estimate the effect size of each factor on each signature by partitioning the total sum of squares ($SS_{tot}$) in the response variable over all combinations of factor levels:

$$SS_{tot} = SS_a + SS_b + SS_c + SS_d + SS_e + SS_I + SS_\varepsilon \qquad (2)$$

where:

$$SS_{tot} = \sum_{i=1}^{n_a}\sum_{j=1}^{n_b}\sum_{k=1}^{n_c}\sum_{l=1}^{n_d}\sum_{m=1}^{n_e}(y_{i,j,k,l,m} - \bar{Y})^2 \qquad (3)$$

where $n_a$, $n_b$, $n_c$, $n_d$ and $n_e$ are the number of levels for each factor, $y$ is the response for a given treatment (i.e. combination of factor levels) and $\bar{Y}$ is the grand mean of the response variable over all treatments. $SS_a$, $SS_b$, $SS_c$, $SS_d$ and $SS_e$ in Eq. 2 are the sum of squares due to the main effects, i.e. the variability in the response variable due to varying a given factor on its own. For example:

$$SS_a = n_b n_c n_d n_e \sum_{i=1}^{n_a}(y_{i,\circ,\circ,\circ,\circ} - \bar{Y})^2 \qquad (4)$$

where $\circ$ indicates averaging over an index. $SS_I$ includes all non-additive interaction terms where the combined effect of two or more factors is not the sum of their main effects. For a 5-factor ANOVA, one could include all unique $n$-tuple combinations of factors where $n = (2, 3, 4, 5)$. Given the difficulty in interpreting these higher-order interactions, and computational requirements, it was decided to investigate the nine first-order interactions only, so that:

$$SS_I = SS_{ab} + SS_{ac} + SS_{ad} + SS_{ae} + SS_{bc} + SS_{bd} + SS_{be} + SS_{cd} + SS_{ce} + SS_{de} \qquad (5)$$

The sum of squares for a first-order interaction are calculated as follows using factors $a$ and $b$ as an example:

$$SS_{ab} = n_c n_d n_e \sum_{i=1}^{n_a}\sum_{j=1}^{n_b}(y_{i,j,\circ,\circ,\circ} - y_{i,\circ,\circ,\circ,\circ} - y_{\circ,j,\circ,\circ,\circ} + \bar{Y})^2 \qquad (6)$$

Finally, the $SS_\varepsilon$ term includes all unexplained variance i.e. error in the ANOVA model.

Having partitioned the sum of squares, the effect size, $\eta^2$ for any term in Eq. 3 can be taken as the proportion of the total sum of squares:

$$\eta_*^2 = SS_*/SS_{tot} \qquad (7)$$

where $*$ can be any of the main effects, interactions or error term.

Bosshard et al. (2013) showed that because ANOVA is based on a biased variance estimator that underestimates the variance in small sample sizes, the calculated effect sizes are biased if a different number of levels are used for each factor. Given that the number of factor levels range from 2 to 24, a pure application of ANOVA using all possible treatments would lead to biased results. Bosshard et al. (2013) outlined a method to correct for this which involves sub-sampling the factor levels down to the smallest number levels across all factors. The procedure is repeated using every possible combination of factor levels with unbiased effect size taken as the mean across all sub-samples. However, given that there are $> 10^8$ unique combinations of factor levels when sub-sampled down to two (and discarding factor level repetitions), it would have been infeasible to account for every possible combination. Instead, it was decided to calculate the effect sizes in this manner using five different sub-sample sizes ($10^1, 10^2...10^5$). The results were then analysed to see if the effect sizes converged. It was found that $10^3$ sub-samples were sufficient to converge the effect sizes for all river discharge signatures and projections of snow and ice coverage. Accordingly, this sub-sampling strategy was adopted in this study.

## 3    Results

### 3.1    Evaluation of calibrated GHM compositions

The simulated river discharge time-series and signatures using the calibrated GHM compositions were evaluated against river discharge observations covering the years 2015 and 2016. Note, no data for mean January and February flows were available for these years. Figures 4a and b show the simulated 'capture ratio' (the ratio of the 336 GHM compositions that capture the observation data within their 95% uncertainty bounds) time-series projected onto the mean observed river discharge for the years 2015 and 2016 respectively. Also shown is the ensemble mean simulated river discharge (black dash) which while not indicative of a single GHM simulation, does provide an indication of overall projection bias.

56% of the observation time-series were captured by at least half of the GHM compositions, while 41% and 28% of the observations were captured by at least 75% and 90% of the GHM compositions. 12% of the observations were not captured by any of the GHM compositions. These included some of the low flows observed at the beginning of the year outside of the melt season, particularly in 2015, where the GHM showed consistent negative biases. Some rainfall-induced summer peak flows were also not captured, particularly during the late summer months of August and September. Furthermore, the sustained summer melt runoff discharge in between rainfall-induced peak flows tended to be overestimated (for example during July and August 2016). Even so, the flow duration curve (FDC) in Fig. 4c shows that almost the entire FDC was captured by all of the GHM simulations except for some of the lowest flows on record. Indeed, Fig. 4d reveals that GHMs were least efficient at capturing the low flow signatures, particularly the variability signature ($\sigma_{99\text{-}95}$), where simulations were positively biased by almost four times the observation uncertainty. For the remaining signatures though, the ensemble of GHMs were remarkably efficient, with the majority of simulations (and in most cases all of them) capturing these signatures within their 95% observation uncertainty bounds.

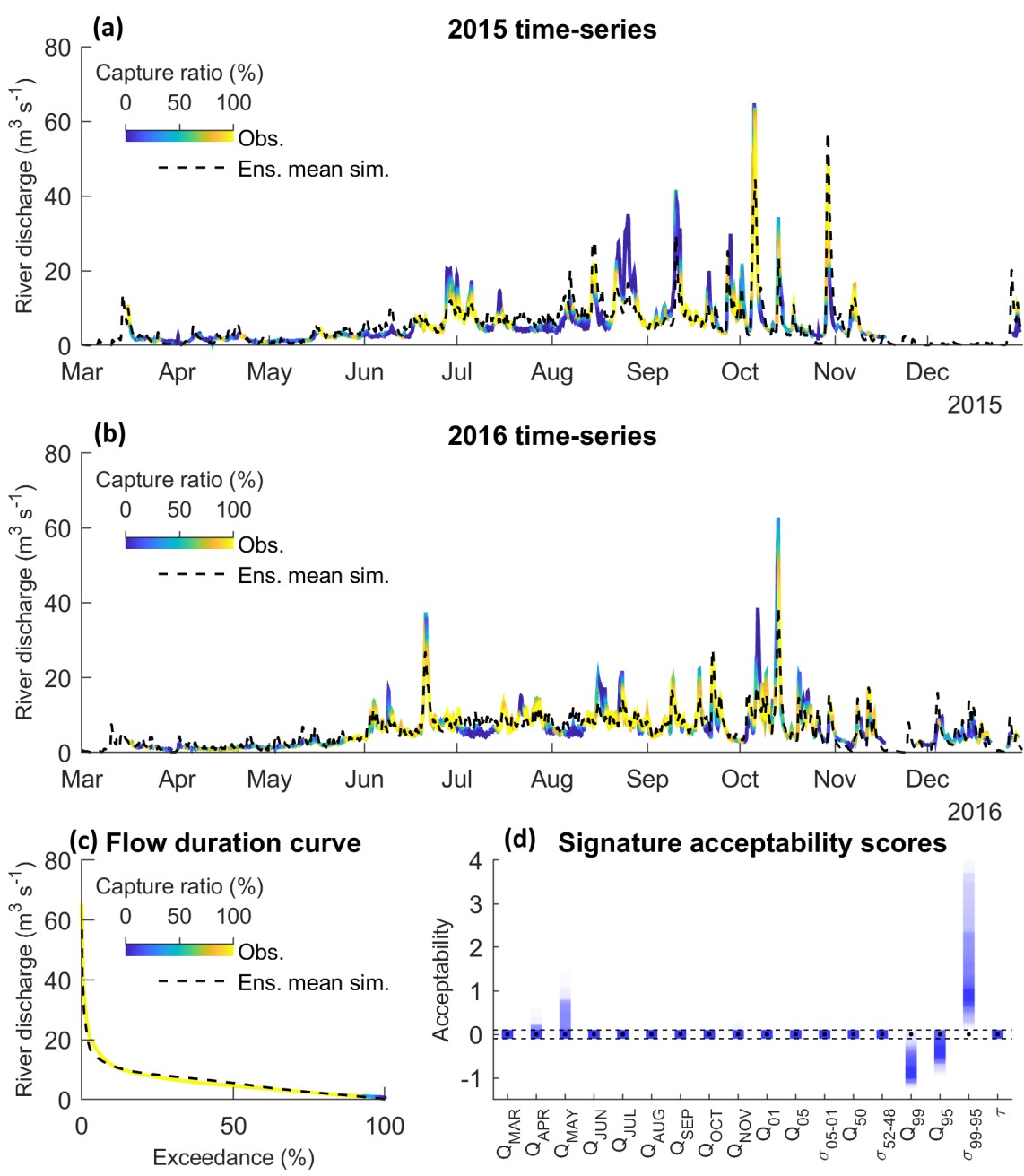

**Figure 4.** Capture ratio projected onto observed river discharge data during evaluation period for 2015 (a); 2016 (b); and over the FDC (c). The weighted ensemble mean simulation is shown as a black dash. Also shown are the range of acceptability scores for each of the available river discharge signatures over the evaluation period (d). Acceptable simulations in (d) are those contained within the black dash lines.

## 3.2 Future climate projections

Projections of temperature for the late 21st century (2076-2100) consistently show an increase relative to the recent past (1981-2005). The largest increases are projected for the coldest days of the year during the winter (Figure 5a), spring (Figure 5d) and autumn (Figure 5j) months as shown by the positive skew in the lower sections of the ECDFs. However, these changes are also typically associated with the greatest projection uncertainty. RCP4.5 projects annual mean near surface air temperature to rise by between 1.1 and 3.6 °C by the late 21st century relative to the recent past with an ensemble mean projection of +2.0 °C. RCP8.5 projects an equivalent rise of between 2.3 and 4.9 °C with an ensemble mean projection of +3.3 °C.

Projected changes in incident solar radiation span positive and negative values, but the median projections are consistently negative indicating reductions in incident solar radiation are most likely. Uncertainty in the magnitude of change is highest during the spring and summer months (Figures 5e and h) when incident solar radiation peaks. Under RCP4.5 annual mean incident solar radiation is projected to change by between -10.7 to +0.8% by the late 21st century with an ensemble mean projection of -4.4%. Under RCP8.5 changes of between -15.3 to 0.4% are projected with an ensemble mean projection of -7.7%.

Projected changes in total precipitation are negligible for the four lowest 10% sections of the precipitation ECDFs, but significant for the two highest sections. In the winter (Figure 5c) and autumn (Figure 5l) months, absolute changes exceed 40 mm d$^{-1}$. The direction of change is uncertain apart from autumn where median projections are consistently positive for the upper sections of the ECDF. The magnitude of change is also uncertain. RCP4.5 projects annual mean precipitation will change by between -13.5 to +21.6% relative to the recent past by the late 21st century with an ensemble mean projection of +1.7%. Under RCP8.5 changes of between -25.7 to 25.1% are projected with an ensemble mean projection of +1.4%.

Figure 6 shows the correlation matrix calculated between seasonal average climate variables for late 21st century. For all climate variables, between-season changes (scores within green borders in Fig. 6) are positively correlated indicating that an increase in summer temperature typically corresponds with an increase in winter temperature for example. Temperature has the greatest between-season correlation while precipitation is the least well correlated. Within-season, between-variable correlation scores (within purple border in Fig. 6) show that precipitation and incident solar radiation are negatively correlated and that the correlation between precipitation and temperature depends on the time of year. For the cooler winter, spring and autumn months, temperature and precipitation are positively correlated, but there is a weak negative correlation for the summer months. Temperature and incident solar radiation are negatively correlated, most strongly for the cooler winter, spring and autumn months.

## 3.3 Future evolution of snow and ice coverage

The ensemble mean projections of annual mean watershed snow coverage (Fig. 7a) show that it will decrease from 12.2 km$^2$ in 2016 to 9.2 km$^2$ in 2100 (25% reduction) under RCP4.5 and 6.0 km$^2$ (51% reduction) under RCP8.5. The 95% projection confidence intervals indicate that by 2100 the watershed could be almost entirely free of snow (2.5 km$^2$ remaining) under RCP8.5 or could have a coverage exceeding present levels (13.3 km$^2$) under RCP4.5.

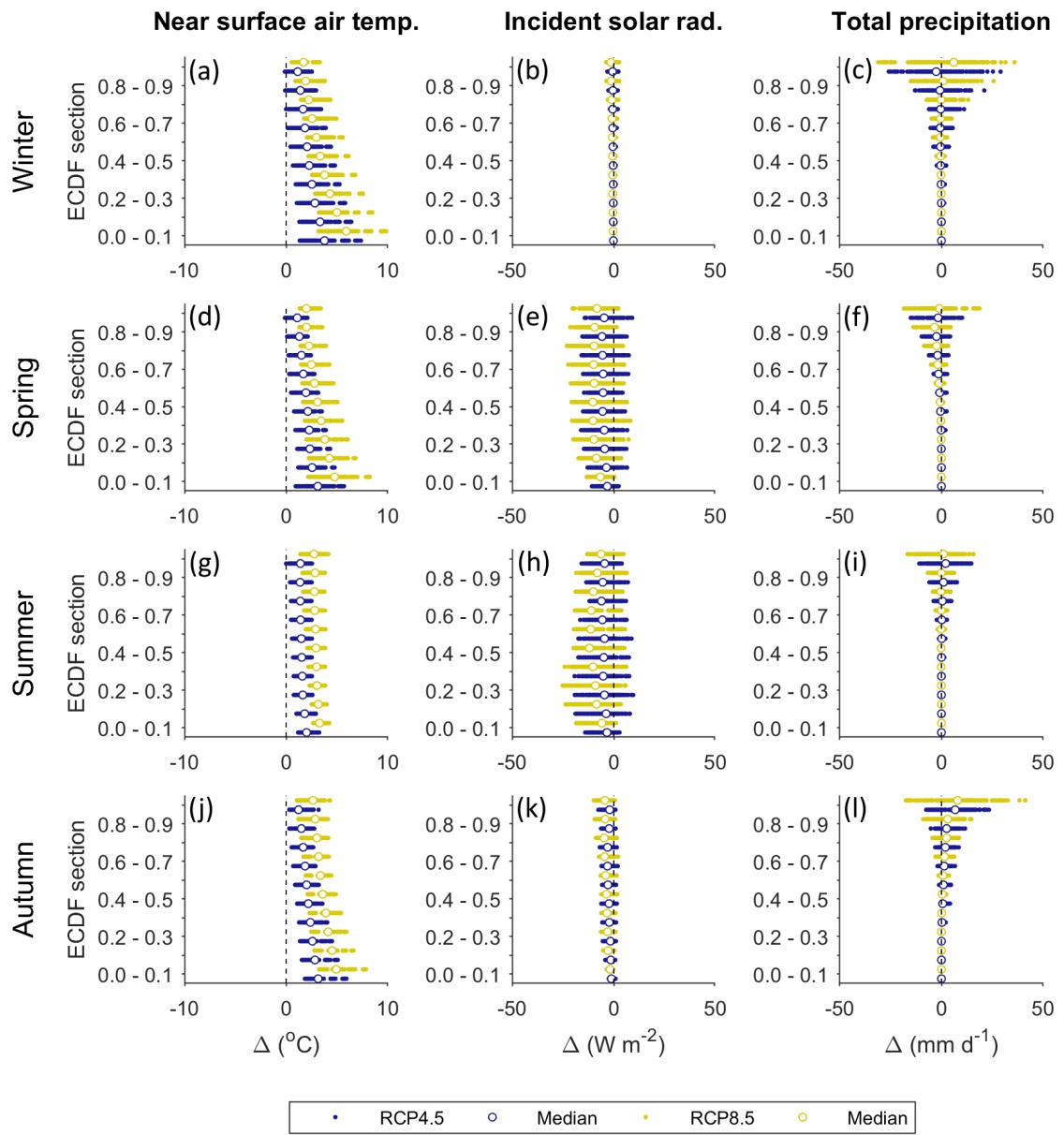

**Figure 5.** Seasonal average projected changes in ECDFs for near surface air temperature (a,d,g,j), incident solar radiation (b,e,h,k) and total precipitation (c,f,i,l) for the late 21st century (2076-2100) relative to the recent past (1981-2005). Changes are plotted for each 10% section of the ECDFs. For each section, blue and yellow dots represent each of the 140 downscaled future climate time-series for the RCP4.5 and RCP8.5 ES respectively (280 in total). Winter = Dec, Jan, Feb; Spring = Mar, Apr, May; Summer = Jun, Jul, Aug; Autumn = Sep, Oct, Nov.

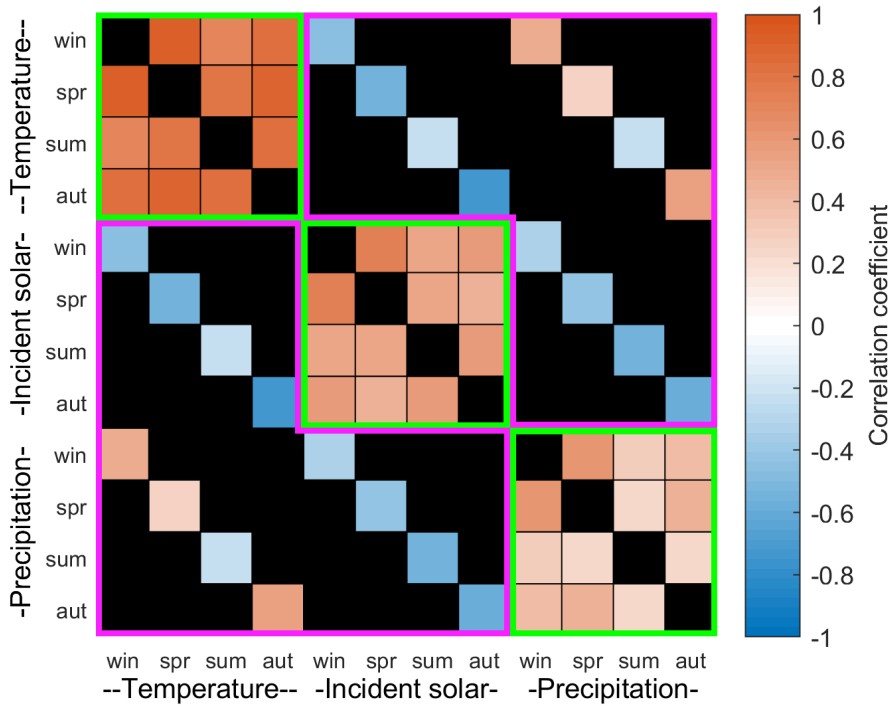

**Figure 6.** Correlation matrix between seasonal average climate variables calculated for late 21st Century (2076-2100) using the 280 down-scaled future climate time-series. Within-variable, between-season correlation scores are contained within the green borders and within-season, between-variable correlation scores are contained within the purple borders. Those regions of the correlation matrix that do not cover these two groups are shaded in black.

Beyond 2050, there is high confidence ($\geq 95\%$) that snow coverage will reduce relative to 2016 levels under RCP8.5 (thin yellow line in Fig. 7a) and equally high levels of confidence apply to projected reductions in snow coverage beyond 2066 under the cooler RCP4.5 (thin blue line in Fig. 7a).

The ensemble mean projection of ice coverage (Figure 7b) projects a 31% reduction relative to 2016 by 2100 under RCP4.5 and a more severe 63% reduction under RCP8.5. There is high confidence ($\geq 95\%$) that ice coverage will be less than 2016 levels from 2037 onwards under RCP4.5 and from 2030 onwards under RCP8.5 but the magnitude of change is uncertain under both emission scenarios. By 2100, the 95% confidence intervals for both RCP4.5 and RCP8.5 are 6.5 km$^2$ wide (more than half the 2016 watershed ice coverage).

The simulation that projected the minimum ice coverage by 2100 under the RCP8.5 emission scenario shows sustained retreat of the glacier between 2000 and 2100 resulting in a watershed that is almost entirely ice free by the end of the century (Figure 8). In contrast, the maximum ice coverage simulation under the RCP4.5 emission scenario projects two periods of

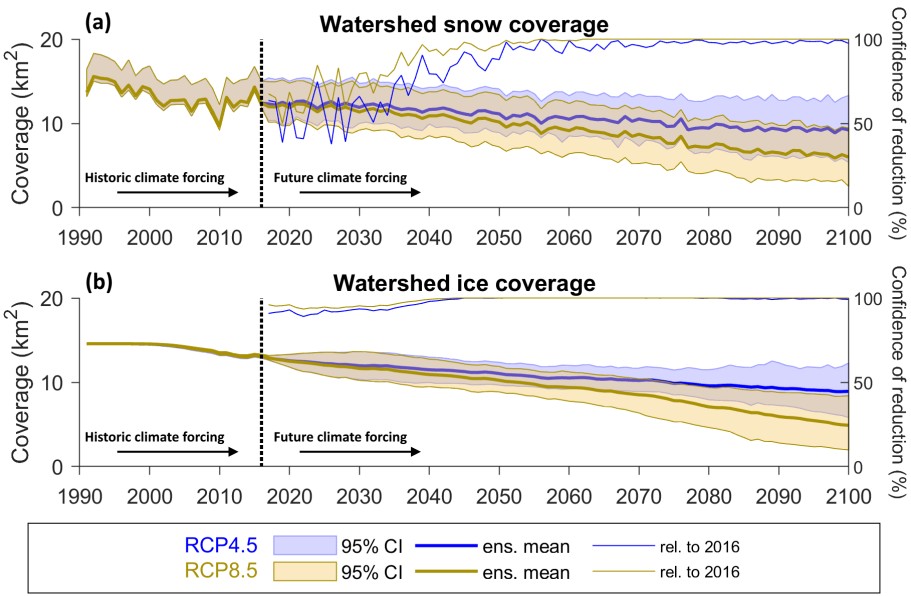

**Figure 7.** Projected annual mean watershed snow coverage (a) and ice coverage (b) including the projection confidence intervals (bands) and ensemble mean projections (thick solid lines) for the RCP4.5 (blue) and RCP8.5 (yellow) projections. Also shown are projection confidence levels for a reduction in coverage relative to 2016 (thin solid lines, right hand axis).

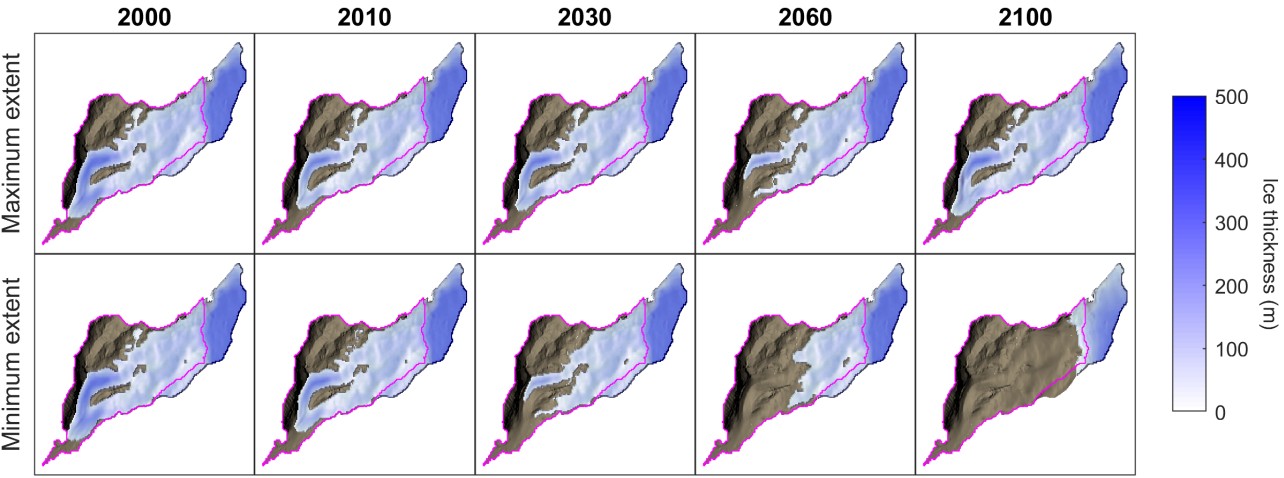

**Figure 8.** Simulated ice thickness between 2000 and 2100 based on simulations that projected the maximum (RCP4.5) and minimum (RCP8.5) ice coverage by 2100. Watershed outline shown in magenta.

glacier advance between 2010 and 2030 and between 2060 and 2100. By the end of the century, this simulation projects ice coverage will be similar to that in 2000.

Figures 9a, b and c show the climate projection time-series that produced the minimum (dotted lines) and maximum (dashed lines) snow (blue lines) and ice (red lines) coverage by 2100. The minimum coverage simulations were forced with some of the highest temperature time-series while the maximum coverage simulations were forced with some of the lowest. The maximum coverage simulations show higher than average incident solar radiation inputs (Figure 9b) and lower precipitation volumes than the minimum coverage simulations. Indeed, correlation scores calculated between seasonal average climate variables and the simulated snow and ice coverage by 2100 (Figure 9d) show that there is a strong negative correlation between mean temperature and projected snow and ice coverage and a weaker positive correlation between snow and ice coverage and incident solar radiation. An even weaker negative correlation exists between autumn and winter precipitation and snow and ice coverage.

## 3.4 Sources of uncertainty in snow and ice coverage projections

The effect size of the main, interaction and error terms calculated using ANOVA for projected changes in snow and ice coverage are shown in Fig. 10. Note, ROR effects are not included here as this model chain component has no influence on cryospheric processes in the GHM. The effect size of each ANOVA term changes through the decades and also varies between snow and ice coverage. Throughout the 21st century, TIM uncertainty contributes <3% to the total projection uncertainty of snow coverage. For projections of ice coverage, $\eta^2_{TIM}$>0.11 up to and including the 2060s, but then gradually falls to 0.07 by the 2080s. $\eta^2_{DS}$ and $\eta^2_I$ never exceed 0.1 for snow and ice coverage and as with $\eta^2_{TIM}$, they gradually reduce through the latter half of the 21st century. GCM-RCM uncertainty is the largest contributor to ice coverage projection uncertainty in the 2030s with an effect size of 0.47. For snow coverage, ES and GCM-RCM have similar effect sizes of 0.45 and 0.4 respectively. However, for the mid and latter half of the 21st century ES uncertainty dominates, contributing 73% and 72% of snow and ice coverage total projection uncertainty by the 2080s respectively.

## 3.5 Future evolution of primary runoff components

As an initial indication of the potential for downstream river flow regime change, Fig. 11 shows the 21st century evolution of changes in the four primary runoff components relative to the reference period. The ensemble means (solid coloured lines) indicate that by the end of the century rainfall will increase for all months under both emission scenarios except for August where RCP8.5 shows a small decrease in rainfall on average (Figure 11a). The largest increases are shown during the autumn (SON) and winter (DJF) months under RCP8.5. The confidence in the direction of change by the end of the century is ≥90% for six months under RCP8.5 (as indicated by the coloured bands), but only for two months (March and April) under RCP4.5. However, ≥75% of the projections from both RCPs project an increase in rainfall between October and April (as indicated by the markers in Figure 11a). A comparison of the reference and 2080s monthly ensemble means (inset in Figure 11a) indicates that the peak rainfall month will shift from September to October.

For snowmelt, the greatest changes are projected to occur in the summer months of July and August under RCP8.5 where there is ≥90% confidence that melt will reduce relative to the reference period from the 2040s onwards (Figure 11b). RCP4.5 also projects decreases in summer melt, but the magnitude of change is smaller. In the winter months, both RCPs project a

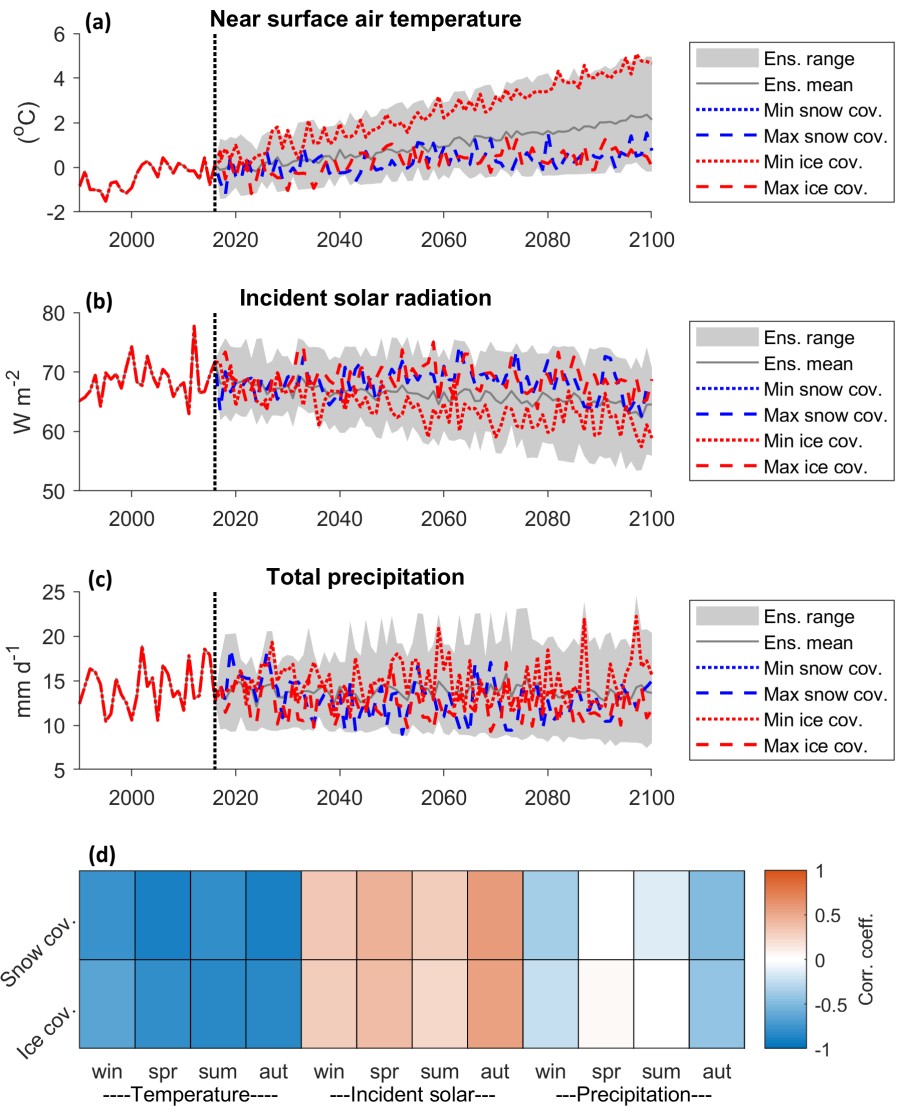

**Figure 9.** Relationship between driving climate data and projected snow and ice coverage including annual mean downscaled climate time-series of temperature (a), incident solar radiation (b) and total precipitation (c) with time-series that produced the minimum (dotted lines) and maximum (dashed lines) snow and ice coverage by the end of 2100. Also included are correlation scores calculated between seasonal average climate variables over the entire future period (2017-2100) and simulated snow and ice coverage by the end of 2100 (d).

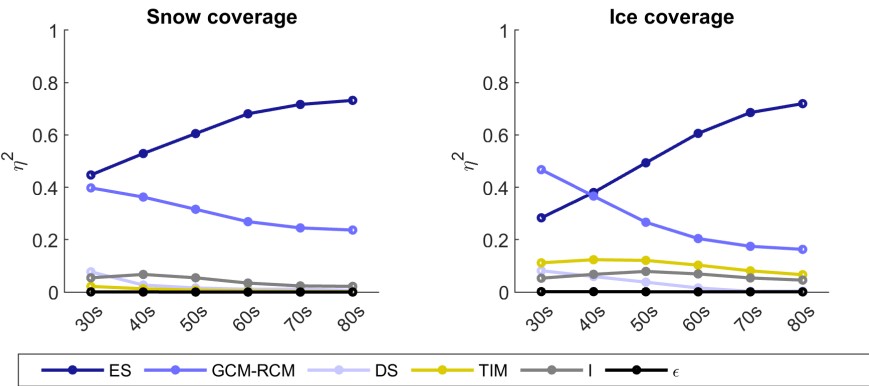

**Figure 10.** Effect size ($\eta^2$) of main effects (ES, GCM-RCM, DS and TIM), interactions (I) and remaining error ($\epsilon$) on projected changes in snow and ice coverage calculated using ANOVA for the six 21st century time-slices. Note, the ROR main effect is not included here as it does not influence cryospheric processes in the GHM.

small increase in melt on average by the end of the century. The ensemble means project that total summer (JJA) melt will reduce by 19% under RCP4.5 and 37% under RCP8.5 by the 2080s (inset in Figure 11b). Annual melt will decrease by 12% under RCP4.5 and 26% under RCP8.5. A similar pattern of change is projected for ice melt (Figure 11c) where total summer (JJA) melt will reduce by 33% under RCP4.5 and 58% under RCP8.5 by the 2080s. There is high confidence ($\geq$90%) that
mean monthly ice melt will reduce for all months except December under RCP8.5. Under RCP4.5 a small increase in winter ice melt is projected for the early and mid 21st century, but by the 2080s, winter melt is projected to reduce near to reference levels on average. Under RCP8.5, winter ice melt is projected to reduce relative to reference levels for the latter half of the 21st century.

Projections consistently ($\geq$90%) show an increase in evapotranspiration for all months of the year (Figure 11d) with the
largest increases projected under RCP8.5 towards the end of the 21st century. However, the volume of evapotranspiration will remain a small component of the overall water balance.

### 3.6    Future evolution of river flow regime

Figure 12 shows the projected changes in river discharge signatures relative to the reference period (except peak water for which the raw projections are shown). Under RCP4.5 the ensemble mean projection of peak water is 2045, while under RCP8.5 peak
water is projected to occur 17 years earlier in 2028. Indeed, the RCP8.5 projections of the mean annual flow signature ($\bar{Q}$) show a consistent decline through the 21st century with $\geq$90% confidence that flows will reduce by the end of the century by 19% on average. In contrast, under RCP4.5 the magnitude of the decline is smaller (ensemble mean projects 7% decrease for 2090s) and the direction of change is more uncertain. Both RCPs project an increase in inter-annual flow range ($R_{ANN}$) throughout the 21st century ($\geq$75% under RCP8.5). Under RCP4.5 the ensemble mean projects a 47% increase in $R_{ANN}$ by the 2080s while
RCP8.5 shows a 71% increase.

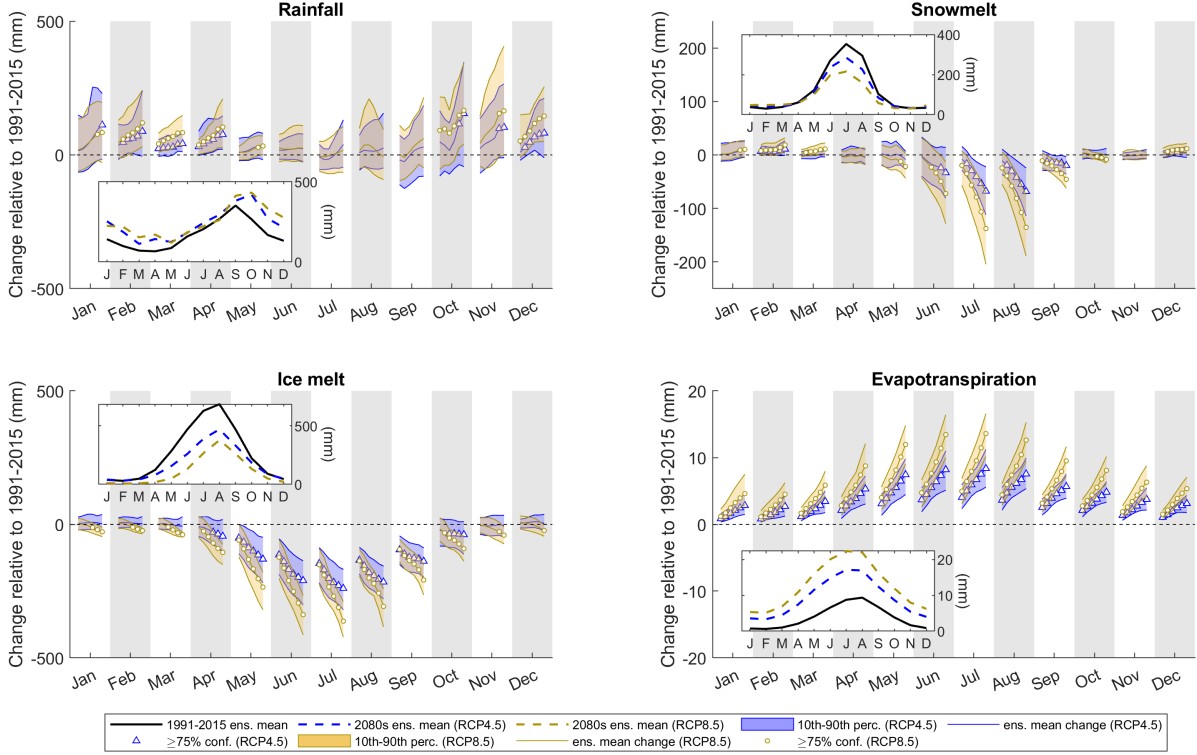

**Figure 11.** Projections of monthly mean runoff components including rainfall (a), snowmelt (b), ice melt (c) and evapotranspiration (d) for RCP4.5 (blue) and RCP8.5 (yellow). For each month, the trajectory of the ensemble mean change over the 21st century time-slices (2030s to 2080s) relative to the reference period (1991-2015) is shown by the solid coloured lines. These lines are marked for each time-slice where there is ≥75% confidence in the direction of change. They are bounded by the 10th and 90th percentiles of the projections (bands). Inset in each plot are ensemble mean monthly runoff volumes averaged over the reference period (black solid line) and 2080s (dashed lines).

Seasonally, monthly winter (DJF) flows are projected to increase while ≥90% of the ensemble project a decrease in summer (JJA) flows by the 2090s under both RCPs. The absolute change in mean monthly flows is larger for summer flows on average, but proportionally, the winter flows are projected to change most, particularly in February where the ensemble mean projects a 60% and 67% increase under RCP4.5 and RCP8.5 respectively by the end of the century. The combined effect of increased

5  winter flows and decreased summer flows results in decreased intra-annual flow variability. Under both RCPs, more than 90% of the ensemble project a decrease in $R_{mnth}$ relative to the reference period from the 2050s onwards. The ensemble mean projections under RCP8.5 show a decade-on-decade reduction in $R_{mnth}$ with time and a 41% reduction by the end of the century.

Of those signatures with units $m^3$ $s^{-1}$, the high flow $Q_{01}$ signature shows the largest ensemble mean increase of 2.8 $m^3$ $s^{-1}$ and 2.5 $m^3$ $s^{-1}$ for RCP4.5 and RCP8.5 respectively by the end of the century. There is high confidence (≥75%) that $Q_{01}$ will

10  increase relative to the reference period under RCP8.5 but the magnitude of change is uncertain under both RCPs. For $Q_{05}$, the ensemble means from both RCPs both show a reduction throughout the 21st century, however the 10th and 90th percentile

span positive and negative values of change for all decades. The ensemble mean projections of changes to high flow variability ($\sigma_{05-01}$) are positive throughout the 21st century under both RCPs. In the latter half of the century, $\geq 75\%$ of the projections under RCP4.5 show an increase in $\sigma_{05-01}$ while $\geq 90\%$ of the projections under RCP8.5 show an increase.

For moderate flows, the ensemble mean of the RCP4.5 projections show a small reduction in $Q_{50}$ of approximately 0.15 m$^3$ s$^{-1}$ throught the 21st century while the RCP8.5 ensemble mean projects a decade-on-decade reduction in $Q_{50}$ and by the end of the century there is high confidence ($\geq 90\%$) that moderate flows will reduce under this emission scenario. Moderate flow variability ($\sigma_{52-48}$) is projected to reduce with high confidence under both RCPs, albeit by only 0.03 m$^3$ s$^{-1}$ and 0.06 m$^3$ s$^{-1}$ by the 2080s under RCP4.5 and RCP8.5 respectively.

For the slow-release low flow signatures, $\geq 90\%$ of the projections are positive throughout the 21st century under both RCPs indicating an increase in the magnitude of low flow events (or equivalently a reduction in the frequency of these flow events) and variability of low flows. The absolute change in the ensemble means never exceed 0.1 m$^3$ s$^{-1}$ for these signatures, although proportionally, they show the largest degree of change, particularly for $Q_{99}$ where the proportional change exceeds 2000% under RCP4.5.

Finally, the response time to runoff ($\tau$) is projected to decrease throughought the 21st century under both RCPs ($\geq 90\%$ confidence) indicating the catchment will likely become more flashy. The magnitude of change is small where the ensemble mean projects a small reduction in $\tau$ of 3.9 hours under RCP4.5 and a slightly greater reduction of 4.7 hours under RCP8.5.

## 3.7 Sources of uncertainty in river flow regime projections

Figure 13 shows the ANOVA effect sizes calculated for the 2030s and 2080s for each river discharge signature. The error term ($\eta_\epsilon^2$) never exceeds 0.09 and for 21 of the 25 signatures is $< 0.03$ indicating that the main effects and first order interaction terms explain the majority of projection uncertainty. For the 2030s, ES uncertainty contributes 4-27% of the total projection uncertainty across the signatures. By the 2080s, ES contributes up to 65% of total projection uncertainty. In fact, for all but four signatures, ES contributes proportionally more to total projection uncertainty in the 2080s than the 2030s. By the 2080s the five signatures with the highest $\eta_{ES}^2$ include the mean monthly flows from May to August and the mean monthly flow range ($R_{mnth}$) signature (Table 2) for which the effect sizes are at least 0.47. GCM-RCM uncertainty is the largest contributor to total projection uncertainty for 21 of the 25 river discharge signatures for the 2030s and it still remains a significant contributor to projection uncertainty by the 2080s with a mean effect size across the signatures of 0.3. Four of the five most sensitive signatures to GCM-RCM uncertainty for the 2030s remain in this top five for the 2080s (Table 2) and these include the mean monthly winter flows in January and February and two of the quick-release high flow signatures ($Q_{01}$ and $Q_{05}$).

On average, the DS parameterisation contributes 18% of the total projection uncertainty across the signatures for the 2030s. In fact, $\eta_{DS}^2$ is relatively consistent across the signatures, ranging from 0.1-0.2 for 18 of the 25 signatures. For the 2080s, $\eta_{DS}^2$ reduces for all signatures except mean November and December flows and the inter-annual flow range ($R_{ANN}$). For $R_{ANN}$, DS has the largest effect size, contributing 43% of the total projection uncertainty. Autumn and winter monthly mean flows for September, November, December and February make up the remainder of the top five signatures most effected by DS uncertainty for the 2080s. On average TIM uncertainty contributes 9% of the total projection uncertainty across the different

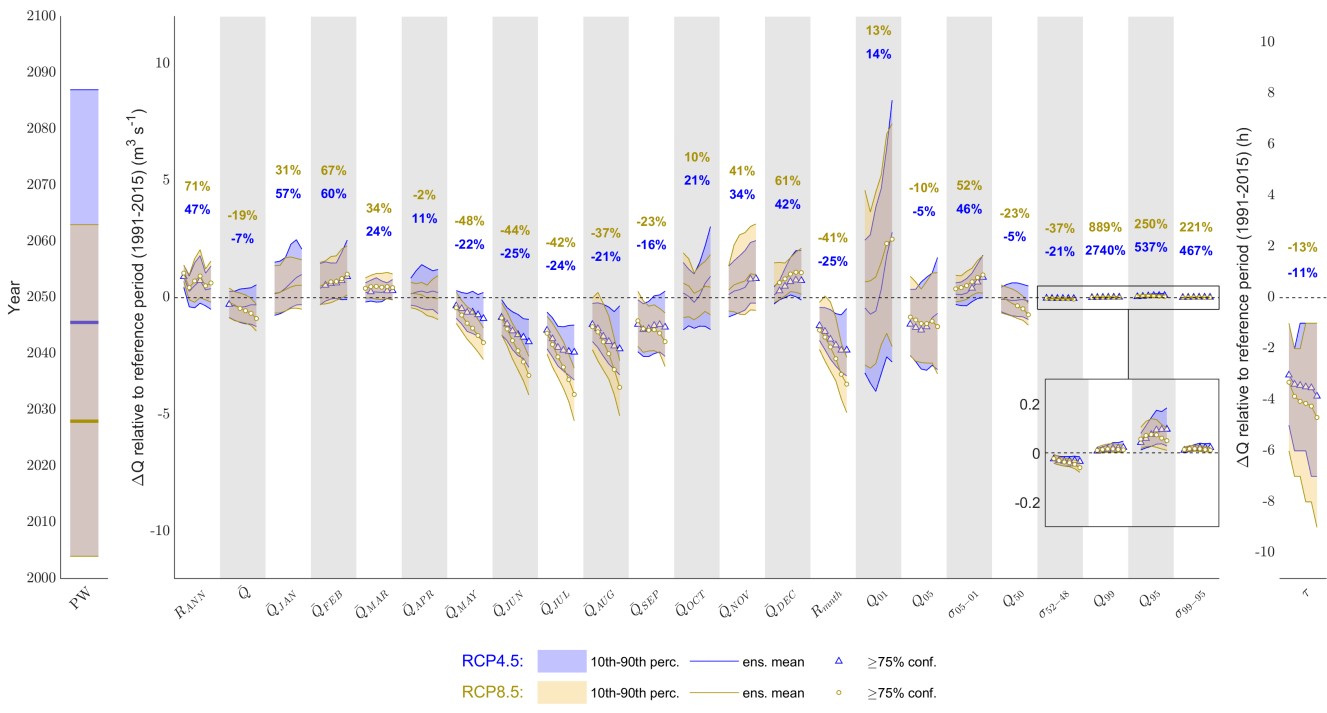

**Figure 12.** Projected changes in river discharge signatures. For each signature, the trajectory of the ensemble mean change over the 21st century time-slices (2030s to 2080s) relative to the reference period (1991-2015) is shown by the solid coloured lines. These lines are marked for each time-slice where there is ≥75% confidence in the direction of change. They are bounded by the 10th and 90th percentiles of the projections (bands). Also shown are 2080s ensemble mean change expressed as a percentage of simulated signatures for the reference period (text). Note, the peak water (PW) signature is not expressed as a change, but as the overall raw projections.

signatures for the 2030s. For this period it is the largest contributor to $R_{ANN}$ uncertainty ($\eta^2_{TIM}$ = 0.35) and it also shows significant contributions to mean monthly flow projection uncertainty for April ($\eta^2_{TIM}$ = 0.17) and May ($\eta^2_{TIM}$ = 0.23) at the beginning of the melt season. It is also the largest contributor to uncertainty of projections of response time to runoff ($\tau$) where $\eta^2_{TIM}$ = 0.20. For the 2080s the average $\eta^2_{TIM}$ falls slightly to 7%, but TIM uncertainty remains an important contributor to

5   total projection uncertainty for $\tau$, April and May flows and two of the low flow signatures ($Q_{95}$ and $\sigma_{99-95}$) where $\eta^2_{TIM} \geq 0.12$. Uncertainty stemming from the ROR structure-parameterisation has a negligible influence on the signatures (PW and $R_{ANN}$) or those signature characterising annual and monthly mean flows for the 2030s and 2080s. For the 2030s it is important for projections of low flow magnitude ($Q_{99}$ and $Q_{95}$, $\eta^2_{ROR}$ = 0.43 and 0.20 respectively) and variability ($\sigma_{99-95}$, $\eta^2_{ROR}$ = 0.13). In fact, for $Q_{99}$, ROR is the single largest contributor to total projection uncertainty. For the 2080s, its influence on low flow

10  quantiles remains significant and it is the single largest contributor to both the $Q_{99}$ and $\tau$ projection uncertainty. It also remains a significant contributor to the high flow variability signature, $\sigma_{05-01}$ where $\eta^2_{ROR}$ = 0.12.

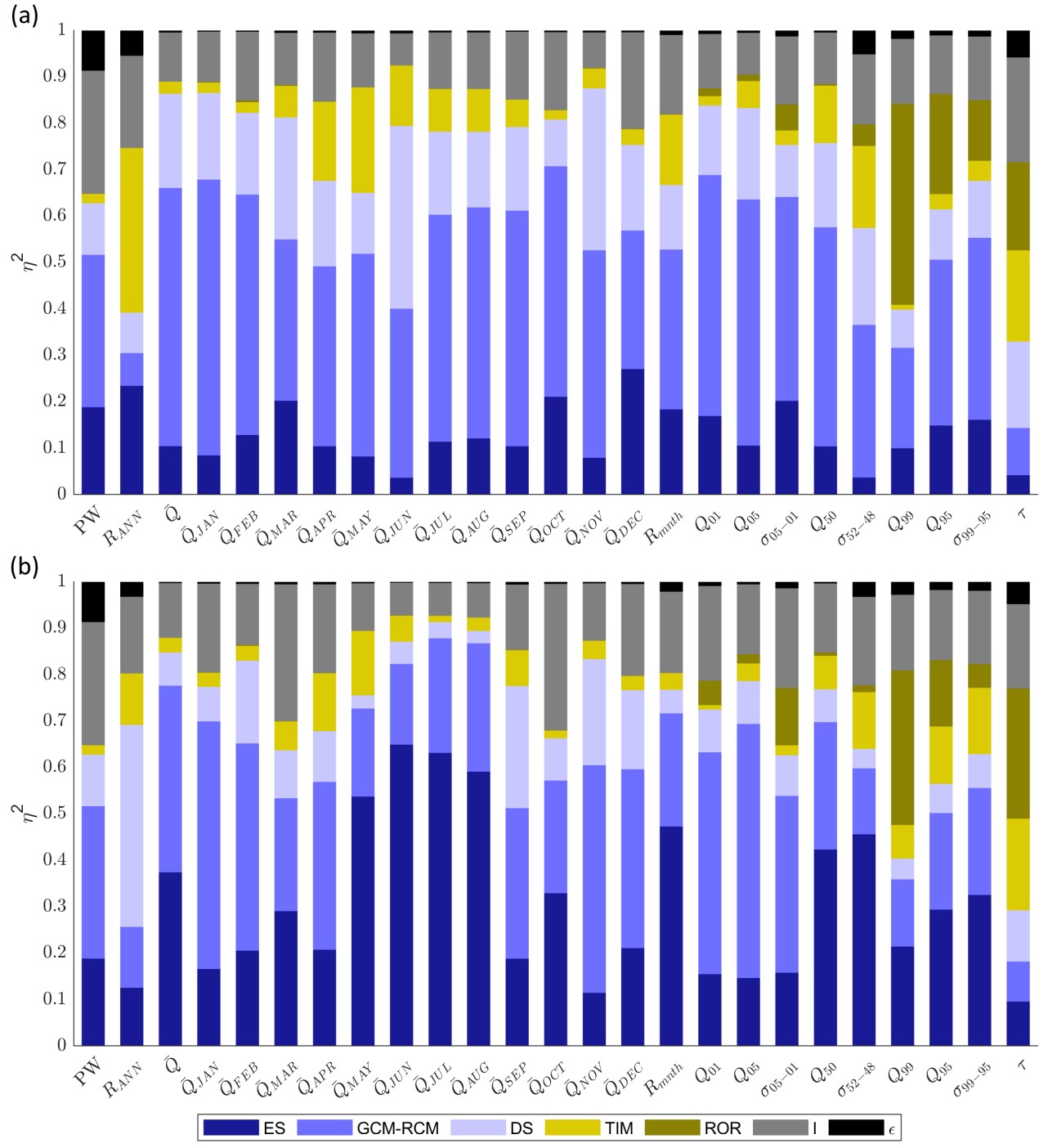

**Figure 13.** Effect size ($\eta^2$) of all main effects (ES, GCM-RCM, DS, TIM and ROR), interactions (I) and remaining error ($\epsilon$) on projected changes in the 25 river discharge signatures at the start (2030s, a) and end (2080s, b) of the 21st century.

**Table 2.** Top five river discharge signatures ranked according to the average effect size for each of the main effects, interactions and remaining error on projected changes for the 2030s and 2080s. Effect sizes are included in brackets.

| Decade | Rank | ES ($\eta^2_{ES}$) | GCM-RCM ($\eta^2_{GCM-RCM}$) | DS ($\eta^2_{DS}$) | TIM ($\eta^2_{TIM}$) | ROR ($\eta^2_{ROR}$) | I ($\eta^2_I$) | $\epsilon$ ($\eta^2_\epsilon$) |
|---|---|---|---|---|---|---|---|---|
| | 1 | $\bar{Q}_{DEC}$ (0.27) | $\bar{Q}_{JAN}$ (0.59) | $\bar{Q}_{JUN}$ (0.39) | $R_{ANN}$ (0.35) | $Q_{99}$ (0.43) | PW (0.27) | PW (0.09) |
| | 2 | $R_{ANN}$ (0.23) | $\bar{Q}$ (0.56) | $\bar{Q}_{NOV}$ (0.35) | $\bar{Q}_{MAY}$ (0.23) | $Q_{95}$ (0.22) | $\tau$ (0.23) | $\tau$ (0.06) |
| 2030s | 3 | $\bar{Q}_{OCT}$ (0.21) | $Q_{05}$ (0.53) | $\bar{Q}_{MAR}$ (0.26) | $\tau$ (0.20) | $\tau$ (0.19) | $\bar{Q}_{DEC}$ (0.21) | $R_{ANN}$ (0.05) |
| | 4 | $\bar{Q}_{MAR}$ (0.20) | $Q_{01}$ (0.52) | $\sigma_{52-48}$ (0.21) | $\sigma_{52-48}$ (0.18) | $\sigma_{99-95}$ (0.13) | $R_{ANN}$ (0.20) | $\sigma_{52-48}$ (0.05) |
| | 5 | $\sigma_{05-01}$ (0.20) | $\bar{Q}_{FEB}$ (0.52) | $\bar{Q}$ (0.20) | $\bar{Q}_{APR}$ (0.17) | $\sigma_{05-01}$ (0.06) | $R_{mnth}$ (0.17) | $Q_{99}$ (0.02) |
| | 1 | $\bar{Q}_{JUN}$ (0.65) | $Q_{05}$ (0.55) | $R_{ANN}$ (0.43) | $\tau$ (0.20) | $Q_{99}$ (0.33) | $\bar{Q}_{OCT}$ (0.32) | PW (0.09) |
| | 2 | $\bar{Q}_{JUL}$ (0.63) | $\bar{Q}_{JAN}$ (0.53) | $\bar{Q}_{SEP}$ (0.26) | $\sigma_{99-95}$ (0.14) | $\tau$ (0.28) | $\bar{Q}_{MAR}$ (0.29) | $\tau$ (0.05) |
| 2080s | 3 | $\bar{Q}_{AUG}$ (0.59) | $\bar{Q}_{NOV}$ (0.49) | $\bar{Q}_{NOV}$ (0.23) | $\bar{Q}_{MAY}$ (0.14) | $Q_{95}$ (0.14) | PW (0.27) | $\sigma_{52-48}$ (0.03) |
| | 4 | $\bar{Q}_{MAY}$ (0.54) | $Q_{01}$ (0.48) | $\bar{Q}_{FEB}$ (0.18) | $\bar{Q}_{APR}$ (0.12) | $\sigma_{05-01}$ (0.12) | $\sigma_{05-01}$ (0.21) | $R_{ANN}$ (0.03) |
| | 5 | $R_{mnth}$ (0.47) | $\bar{Q}_{FEB}$ (0.45) | $\bar{Q}_{DEC}$ (0.17) | $Q_{95}$ (0.12) | $Q_{01}$ (0.05) | $Q_{01}$ (0.20) | $Q_{99}$ (0.03) |

Unlike ice and snow coverage, interactions between model components significantly contribute the total projection uncertainty across the signatures where $\eta^2_I$ ranges between 0.07 and 0.27 for the 2030s and between 0.07 and 0.32 for the 2080s. Figure 14 shows the decomposition of the five interaction terms with the largest effect sizes on average for the 2030s (a) and 2080s (b). The interaction between the ES and GCM-RCM model chain components dominate the contribution to projection uncertainty. However, interactions between the climate model chain components and the GHM (e.g. DS-TIM) may also contribute to the projection uncertainty. For $R_{ANN}$, DS-TIM interaction contributes 7% of total projection uncertainty for the 2030s and 2080s. Furthermore interactions between the TIM and ROR in the GHM contribute some (albeit small) amounts to the total projection uncertainty. For 16 of the signatures, the contribution from interactions between model chain components increases from the 2030s to the 2080s. These include all of the signatures that characterise, high, moderate and low flow magnitude and variability, but the largest increases are shown for March and October mean monthly flows.

## 4 Discussion

### 4.1 Future evolution of river flow regime

There is high confidence that near-surface air temperature will increase by the late 21st century (2076-2100) relative to conditions in the recent past (1981-2005). Precipitation and incident solar radiation were projected to slightly increase and decrease respectively on average: a finding that is consistent with other analyses of the EURO-CORDEX projections for northern Europe (Bartók et al., 2017). The primary driver of changes in snow and ice is near-surface air temperature, while precipitation and incident solar radiation are of secondary importance. Because of this, there is high confidence that glacier ice and snow will continue to retreat as near-surface air temperature rises throughout the 21st century which would leave the river basin almost free of snow and ice by 2100 under the warmest climate projections.

The signature-based analysis undertaken in this study has revealed how climate change will impact the magnitude, timing and variability of downstream river flows over a range of timescales in the Virkisá river basin. Projected changes in flow seasonality broadly follow those shown for other mid-latitude alpine river basins where the loss of snow and ice will reduce

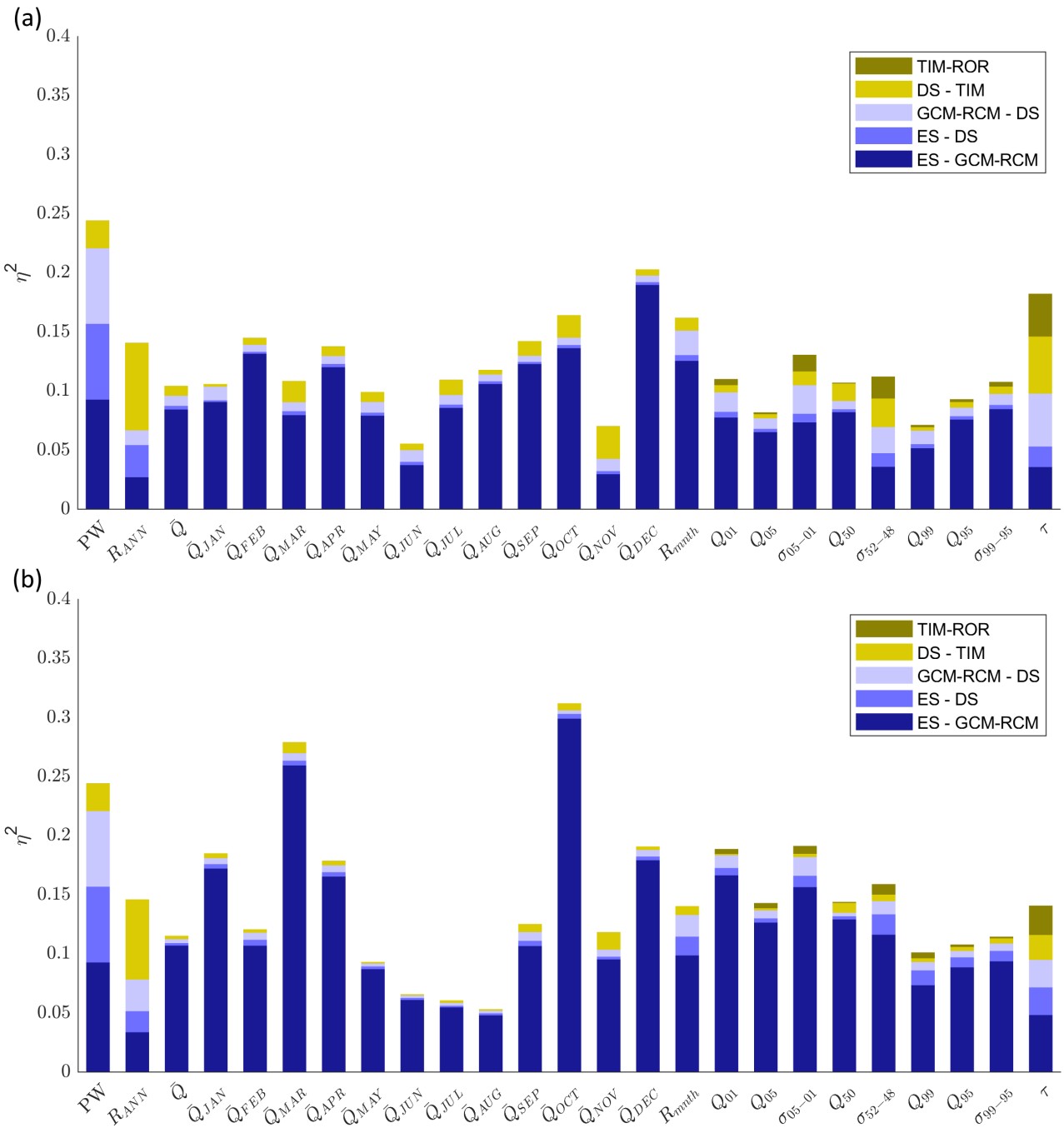

**Figure 14.** Effect size ($\eta^2$) of the five most significant interactions on projected changes in the 25 river discharge signatures at the start (2030s, a) and end (2080s, b) of the 21st century.

meltwater inputs in the summer months and a phase shift of precipitation from snowfall to rainfall combined with enhanced melt during the colder months will increase winter runoff (Addor et al., 2014; Huss et al., 2014; Mandal and Simonovic, 2017; Jobst et al., 2018). Summer runoff is projected to decrease by 24% under RCP4.5 and 40% under RCP8.5 by the 2080s while winter runoff is projected to increase by 59% under RCP4.5 and 57% under RCP8.5 by the 2080s. The consequence of these seasonal shifts in runoff is that intra-annual (monthly) flow variability will reduce by 25% (RCP4.5) and 41% (RCP8.5) by the 2080s. Furthermore, the magnitude of very low flow events ($Q_{99}$), which typically occur during the winter months, are likely to increase.

On average, the projections indicated that the seasonal redistribution of runoff will have little influence on mean annual flows under RCP4.5 (-7% by the 2080s) as changes in summer and winter flows approximately compensate one-another. Under RCP8.5, however, the more pronounced reduction in summer melt inputs results in a 19% reduction by the 2080s. The loss of a consistent melt input to the river basin and its evolution to a hydrological regime governed by rainfall-runoff processes means inter-annual flow variability ($R_{ANN}$) will increase by 47% (RCP4.5) and 71% (RCP8.5) by the 2080s. The increase in rainfall inputs, particularly during the storm-prone autumn and winter months, likely explains the projected increased magnitude of very high flow events ($Q_{01}$), a finding that is in agreement with other studies that have investigated changes in high flow magnitudes in glaciated river basins (Lutz et al., 2016). It is likely that the intensification of peak flow magnitudes will be further exacerbated by the projected decrease in river flow response time to runoff ($\tau$): an artefact of losing the runoff-regulating ice and snow water stores. Accordingly, the river basin will become more flashy and flood-prone in the future.

Increased flood frequency has major implications to local infrastructure in the vicinity of the Virkisá river basin. In particular, the southern section of the Route 1 highway which passes over the Skeiðarársandur floodplain navigates a large number of glacier-fed rivers including the Virkisá. Due to the unconsolidated nature of the floodplain lithology, the morphology of these rivers can change rapidly, particularly during high flow events (Marren, 2005) and often at considerable expense to the road authority (Björnsson and Pálsson, 2008). Accordingly, the projected increase in frequency and severity of high flow events will likely incur further expenses to maintain this transport link in the future.

Beyond local implications, one should be cautious in extrapolating the findings from this study to other glaciated catchments in Iceland or beyond, but it is clear that the timing, magnitude and variability of glacier-fed river flows over a range of timescales are sensitive to climate change. For Iceland, these changes could impact glacier-fed hydroelectric dams: a primary source of electricity for the country. Increased frequency and magnitude of high flow events could render current dams unsafe if their designed flood capacity can no longer meet regulation requirements (Thorsteinsson and Björnsson, 2012). The redistribution and 'levelling out' of seasonal flows, however, could actually have a beneficial effects on the running costs and capacity to produce electricity from such projects (Jóhannesson et al., 2007).

## 4.2 Uncertainties in projections of river flow regime

Projections of the direction of change relative to the reference period were well constrained for the majority of river discharge signatures, particularly towards the end of the 21st century and for the warmer RCP8.5 emission scenario. Even so, there was considerable spread in the projected magnitude of these changes due to uncertainties in the driving climate data (ES,GCM-

RCM,DS) and representation of glacio-hydrological processes (TIM,ROR) in the model chain. Uncertainty in future snow and ice coverage primarily stemmed from the ES due to it's control on future near-surface air temperature. In fact, the proportional contribution of the ES to projection uncertainties increased throughout the 21st century and consequently, the ES was also found to be the dominant source of uncertainty for projections of mean monthly flows during the melt season by the 2080s.

The growing influence of the ES over time was also shown by Addor et al. (2014) for six alpine catchments in Switzerland and by Duethmann et al. (2016) for two mountain river basins in the Tian Shan. Interestingly though, these studies along with the recent study of Jobst et al. (2018) found that climate model uncertainty was still the dominant source for projections of monthly river flows. Jobst et al. (2018) postulated that this was likely because of the high uncertainty in future precipitation across the climate models. Indeed, others have also attributed future runoff uncertainty in glaciated river basins to variability

in precipitation projections (Lutz et al., 2016), a finding which is compounded by an increasingly warm and thus rainfall-dominated precipitation input. In this study however, the GCM-RCM model chain component only dominated river flow projection uncertainty during the winter months while summer flow uncertainty was dominated by the ES. There are two key reasons that could explain this. Firstly, precipitation uncertainty across the GCM-RCMs showed to be especially high during winter (Fig. 5) which coupled with the fact that rainfall is the primary source of runoff during winter, likley explains the

dominant role GCM-RCM plays in projection uncertainty during the winter months. Furthermore, it should be noted that the Virkisá river basin has a much higher proportional glacier coverage (60%) compared to the aforementioned studies (1.8%-22.3%). Therefore, it is postulated that the influence of the ES in the summer is related to the relatively high proportion of melt runoff that the Virkisá river receives during these months and the fact that the ES showed to be the dominant contributor to future ice coverage uncertainty. Importantly, this finding also serves to highlight the need to represent atmosphere-cryosphere-

hydrosphere feedbacks adequately in future studies, particularly where glacier coverage is high, through the inclusion of a dynamic glacier evolution model in the model chain like that implemented in this study.

For projections of the inter-annual flow range, the DS procedure was the largest contributor to projection uncertainty by the end of the 21st century, which should be expected given that the perturbation of this procedure accounted for uncertainty in the random year-by-year sampling of the historic climate data. Uncertainty in the TIM structure-parameterisation was the dominant

contributor to the spread in projections of moderate monthly flows during the transition from the cold to melt season which corroborates with the model comparison study of Mackay et al. (2018) who found that the structural representation of melt was important for controlling the initiation of the melt season due to the contrasting sensitivity of the models to temperature and incident solar radiation. Mackay et al. (2018) also concluded that signatures derived from the flow duration curve as well as those representing flashiness were most sensitive to the configuration of the ROR component of the GHM. Indeed, here it

was found that uncertainty in the ROR structure-parameterisation significantly contributed to the total projection uncertainty of slow-release low flow signatures as well as the response time (flashiness) of the catchment to runoff. Similar sensitivities in low flow metrics to the choice of hydrological model have been shown for non-glaciated river basins Yuan et al. (2017) and they postulated that these might stem from differences in water storage-release processes in the models. However, a key drawback of this study and other studies that have investigated the role of hydrological model uncertainty in glaciated river

basins (e.g. Giuntoli et al., 2015; Vetter et al., 2017) is that they have implemented multiple model codes and therefore cannot

make any definite conclusion about the source of the projection uncertainties. For example, Addor et al. (2014), concludes that the sensitivity to the choice of hydrological model could stem from any number of differences between model codes including the structure, climate interpolation method and calibration strategy. In this study, it has been demonstrated that by using a single but flexible model code, it is possible to separate out the sources of projection uncertainties down to the process level. Such insights can be used to help prioritise those aspects of the GHM that require: i) additional refinement (e.g. through model development); and ii) adequate representation of their uncertainty to improve projection robustness.

Furthermore, the signature-based analysis undertaken in this study has shown that the importance of these different sources, be it from the GHM or the climate projections is dependent on which signature of river discharge is being evaluated. It is clear, therefore, that signature-based analyses could be used to help prioritise uncertainty sources based on the characteristic of flow one is interested in. For example, the results from this study indicate that for evaluating changes in monthly melt season runoff only, it may be beneficial to ignore ROR uncertainty and focus time and computational resources on quantifying uncertainties stemming from the remaining model components. In this respect, the time frame of the projections should also be considered, given the apparent change in effect sizes with time demonstrated for projections of snow and ice coverage and river flow signatures (see Appendix D).

More broadly, the results from this study emphasise the need for impact studies to represent uncertainties stemming from model chain components that control future climate and glacio-hydrological behaviour, the second of which has been widely neglected. The need for this is compounded by the fact that interactions between model chain components exceeded individual main effects for some river discharge signatures. Accordingly, an ensemble that includes perturbations of multiple components of the model chain simultaneously will provide the most rigorous quantification of projection uncertainty.

## 4.3 Limitations

While some characteristics of projected river flow regime change are broadly in agreement with other studies in similar mid-latitude alpine settings (e.g. changes in flow seasonality and projected increase in high flow magnitude), it is important to emphasise that the projected river flow regime shifts should not be generalised across glaciated river basins. Indeed, recent regional (Ragettli et al., 2016) and global (Huss and Hock, 2018) studies have shown that local catchment characteristics such as climate and glacier hypsometry largely influence seasonal river flow response to 21st century climate change. In this study a small absolute increase in low flow magnitude was projected indicating climate change and deglaciation could help to limit periods of water scarcity. However, in more arid regions, where rainfall cannot compensate reductions in melt, the opposite effect has been shown (Stewart et al., 2015). One might also expect to see much greater changes in the river flow response time to runoff as snow and ice retreat in other river basins. For the Virkisá river basin, a relatively small reduction in response time ($\tau$) was projected on average by the end of the 21st century. This, perhaps, should not be surprising given the small size of the river basin and the fact that previous investigations have shown that Virkisjökull has a well developed conduit drainage system that routes runoff efficiently year-round (Phillips et al., 2014; Flett et al., 2017). For larger river basins with more expansive cryospheric water stores, changes in the response time to runoff could be much greater, substantially increasing the risk of pluvial flooding.

Similar inter-catchment variability should also be expected with regards to the dominant sources of projection uncertainty. Indeed, as already noted in this discussion, some of the results from this study contrast the limited number of studies that have investigated uncertainty sources in other glaciated basins. Addor et al. (2014) suggests that catchment elevation influences the importance of the ES on projection uncertainty whereby runoff from higher elevation catchments with more snow and ice are more sensitive to the ES. It is therefore vital that signature-based evaluations like the one undertaken in this study are applied to other glaciated river basins in the future so that regional variations in river flow regime change and uncertainty sources can be evaluated.

It is also important to consider potential deficiencies in the calibrated GHMs. In fact, the model evaluation demonstrated that they were able to capture the majority of the observed river discharge signatures within their observation uncertainty bounds. Even so, it should be noted that there are several limitations in the calibration approach that could have hindered the efficiency of the calibrated models. Firstly, given the distributed structure of the GHM and the fact that it runs on an hourly time-step, running the GHM over multiple years required considerable computation time which limited the number of runs that could be undertaken in the Monte-Carlo calibration procedure. 5000 runs was adopted as an appropriate compromise, balancing the density of parameter sampling with available computational resources. Even so, it is recognised that particularly for the more complex model structures which employ more calibration parameters, a denser parameter sampling could help to find more efficient model parameterisations. It should also be noted that the models were calibrated and evaluated on four years of river flow data only. This detail is particularly important given the conceptual nature of the GHM and thus the potential for the calibration parameters to become less applicable when applied to periods outside of the calibration data. Additionally, it is important to highlight possible model deficiencies brought about by the two-step GHM calibration procedure employed in which the TIM and ROR model chain components were calibrated independently. This was necessary so that the main effects (Eq. 4) and interaction terms (Eq. 6) for both components could be calculated separately (thus achieving the second aim of the study). However, the drawback of implementing this step-wise calibration procedure over one that calibrates both model components simultaneously is that it neglects any interactions between the TIM and ROR models. Of course, its should be noted that the ANOVA results showed that TIM and ROR interactions are negligible except for two of the 25 signatures evaluated.

In the previous model evaluation study undertaken by Mackay et al. (2018), they highlighted the historic observed precipitation data as source of model deficiencies. They noted the lack of available precipitation data at higher elevations, making the gridded dataset employed in this study less reliable near the basin summit. They also analysed the effectiveness of the bias-correction procedure applied to the precipitation dataset and showed that it resulted in time-series that were well correlated to the AWS data over a 3-day time step, but that this correlation degraded at shorter daily and hourly time steps which could have contributed to the model's inability to capture snow coverage observations higher up in the catchment and river discharge signatures relating to the timing of flows.

Indeed, uncertainties in the historic precipitation data were not included as part of this study, partly because there was almost no information that would have allowed one to quantify these uncertainties (e.g. rain gauge errors), particularly higher up in the catchment where measurements are least reliable. Additionally, though, it would have meant further increasing the size

of the model chain ensemble which was already at the very limit of what was computationally feasible. This, however, raises an important broader limitation of the study in that the total projection uncertainties reported are not indicative of the 'true' uncertainty. Further insights could undoubtedly be gained by perturbing other model chain components including the historic climate time-series and components related to key glacio-hydrological processes such as the snow redistribution routine and

glacier evolution model. Certainly, Jobst et al. (2018) calculated that the bias-correction of precipitation contributed up to 22% of seasonal streamflow projection uncertainty.

Furthermore, the representation of uncertainty in the five components evaluated in this study are themselves not exhaustive. It is well established that uncertainties in climate model ensembles are under-represented (Daron and Stainforth, 2013) and steps were taken in this study to limit the total ensemble size so that the experiments were computationally feasible. For

example, only 10 random DS sequences were generated, and indeed other aspects of the downscaling procedure could have also been modified (e.g. replacing the linear interpolation of change factors with a moving average model). Additionally, the melt and runoff-routing model structures implemented represent a sub-set of a much larger population of available models. For example, we adopted simplified energy balance models and the concept of linear reservoirs to route runoff. However, other model structures that employ more complex physically-based energy balance approaches and hydraulic models that simulate

discrete flow pathways through the glacier (e.g. Arnold et al. 1998) could also be implemented to provide a more accurate representation of the 'true' projection uncertainty.

## 5   Conclusions

21st century climate change is projected to alter the magnitude, timing and variability of river flows over decadal to sub-daily timescales in the Virkisá river basin. Relative to the 1990s reference period, there was high confidence in the direction

of change for the majority of the 25 river discharge signatures over the 21st century. The magnitude of change, however, was more uncertain. The application of ANOVA demonstrated that the climate model chain components (ES,GCM-RCM,DS) were the main sources of this uncertainty. However, uncertainty relating to glacio-hydrological process representation in the model chain (TIM,ROR) were the dominant source of projection uncertainty for some river discharge signatures. Furthermore, interactions between model chain components can exceed individual main effects. Based on these findings, we make several

recommendations for future studies that aim to assess climate change impact on glacier-fed river flows:

1. Studies should seek to evaluate multiple characteristics of river flow regime change (magnitude, timing and variability) over different timescales where possible so that a more thorough understanding of potential environmental and socio-economic impacts can be deduced from projections. Signatures of river discharge provide the ideal tool to evaluate these changes quantitatively. Changes in the magnitude of river flows over decadal to seasonal timescales are already known to

be highly site-specific and therefore we should expect that other signatures of regime change will also show considerable inter-catchment variation.

**Table A1.** List of GCMs and RCMs used in this study.

| Model name | Institution | Type | Driving GCMs |
| --- | --- | --- | --- |
| CNRM-CM5 | National Centre for Meteorological Research | GCM | - |
| EC-EARTH | Europe-wide consortium | GCM | - |
| IPSL-CM5A-MR | Institut Pierre-Simon Laplace | GCM | - |
| HadGEM2-ES | Met Office Hadley Centre | GCM | - |
| MPI-ESM-LR | Max Planck Institute for Meteorology | GCM | - |
| NorESM1-M | Norwegian Climate Center | GCM | - |
| CCLM4-8-17 | Climate Limited-area Modelling Community | RCM | CNRM-CM5, EC-EARTH, HadGEM2-ES, MPI-ESM-LR |
| ALADIN53 | National Centre for Meteorological Research | RCM | CNRM-CM5 |
| RCA4 | Swedish Meteorological and Hydrological Institute, Rossby Centre | RCM | CNRM-CM5, EC-EARTH, HadGEM2-ES, MPI-ESM-LR |
| HIRHAM5 | Danish Meteorological Institute | RCM | EC-EARTH, NorESM1-M |
| RACMO22E | Royal Netherlands Meteorological Institute | RCM | EC-EARTH, HadGEM2-ES |
| WRF331F | Institut Pierre Simon Laplace and Institut National de l Environnement industriel et des RISques | RCM | IPSL-CM5A-MR |
| REMO2009 | Helmholtz-Zentrum Geesthacht, Climate Service Center, Max Planck Institute for Meteorology | RCM | MPI-ESM-LR |

2. Studies should account for uncertainties stemming from both the climate projections and glacio-hydrological process representations so that more robust projections of river flow regime change are produced. The latter has largely been neglected in studies to date.

3. Careful consideration of which model chain components are the dominant sources of projection uncertainty (through the use of methods such as ANOVA) would help to prioritise resources (e.g. computational) to further enhance projection robustness. The results from this study indicate that such decisions will depend on the signatures of river flow regime change that one is interested in projecting.

## Appendix A: EURO-CORDEX models

A total 15 unique GCM-RCMs using six GCMs and seven RCMs were available to use in this study (Table A1). Figure A1 shows the EURO-CORDEX 0.11° RCM grids. After comparing monthly average simulations from each GCM-RCM over the recent past (1981-2005) against the observed climate data, it was found that the [CNRM-CM5]-[ALADIN53] GCM-RCM has anomalously large negative temperature biases, particularly during the winter months of the year (see red line in Fig. A2d). In addition to this, a root mean squared error (RMSE) score was calculated for each climate variable by comparing monthly observed and simulated empirical distribution functions constructed from catchment-average daily climate data (Fig. A2a-c). When ranked according to their RMSE scores, the [CNRM-CM5]-[ALADIN53] GCM-RCM ranked 14, 13 and 15 out of 15. Given the anomalously high biases in temperature and the importance of temperature for driving hydrological change in the catchment (both in terms of melt rate and the proportion of precipitation falling as rainfall), coupled with the fact that the model was relatively poor across all three climate variables, it was deemed appropriate to remove this model from the ensemble.

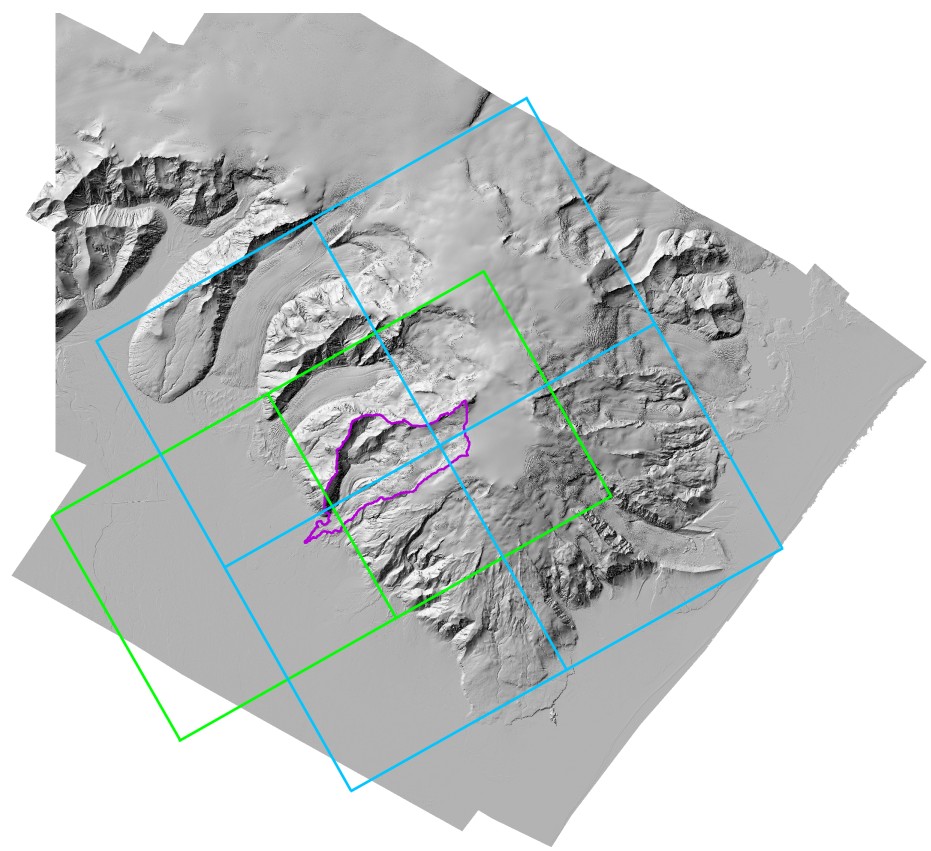

**Figure A1.** EURO-CORDEX 0.11° RCM grid lines. RCM nodes are situated at grid line intersects. All RCMs utilise the green grid except for REMO2009 which uses the blue grid.

## Appendix B: Ice melt and snow coverage signatures used for model calibration

12 signatures of ice melt and snow coverage which were previously derived by Mackay et al. (2018) were used for model calibration and are shown in Table B1. These signatures include: i) measurements of winter and summer ice melt in the main ablation zone between 2012 and 2014 which were derived from ablation stake data; ii) an estimate of long-term glacier volume change calculated using two DEMs of the ice for 1988 and 2011; and iii) estimates of the average seasonal snow coverage for spring (March and April) early summer (May and June) and late summer (July and August) in the lower (77 - 587 m asl), middle (587 - 776 m asl) and upper (776 - 1123 m asl) sections of the glacier-free basin area. These were calculated from the remotely-sensed MOD10A1 MODIS product for the years 2001 to 2015 inclusive (Riggs and Hall, 2015).

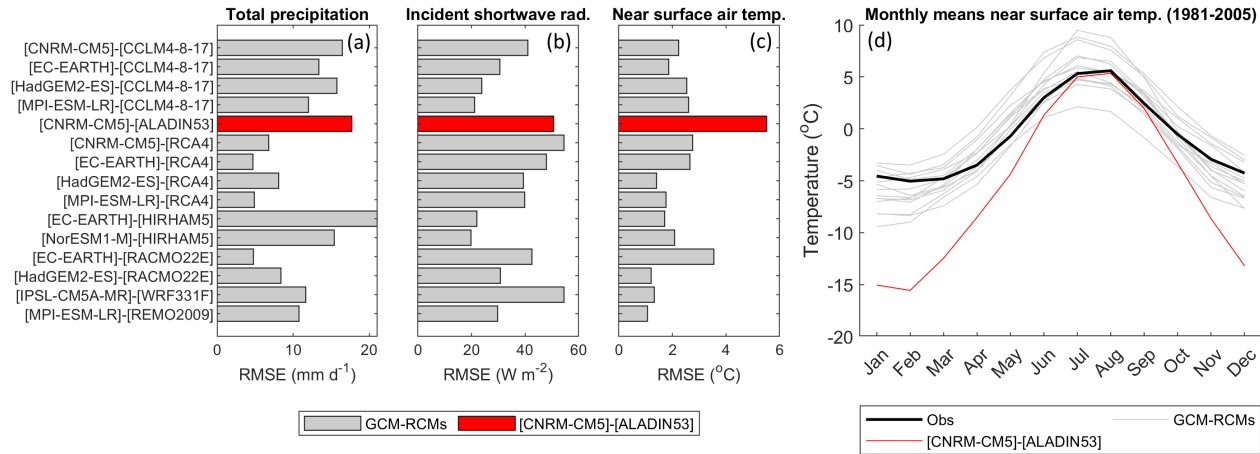

**Figure A2.** Root mean squared error scores calculated by comparing monthly empirical distribution functions constructed from catchment-average daily observed and simulated (GCM-RCM) total precipitation (a), incident solar radiation (b) and near surface air temperature (c) data over the recent past (1981-2005). Also shown are the observed and simulated monthly mean near surface air temperatures for the recent past (d).

**Table B1.** Summary of 12 ice melt and snow coverage signatures used to calibrate the GHM with their limits of acceptability. Note, snow coverage is expressed as a proportion of the glacier-free basin area.

| Attribute | Signature | Limits of acceptability |
|---|---|---|
| Seasonal ice melt on tongue | 2013 Summer ice melt | 5.22 – 6.44 m we |
| | 2012-2013 Winter ice melt | 0.64 – 0.78 m we |
| Long term glacier volume change | Change in ice volume (1988-2011) | -0.36 – -0.28 km$^3$ |
| Snow coverage in lower catchment | Mean snow coverage in spring | 0.32 – 0.45 |
| | Mean snow coverage in early summer | 0.02 – 0.08 |
| | Mean snow coverage in late summer | 0.00 – 0.03 |
| Snow coverage in mid catchment | Mean snow coverage in spring | 0.70 – 0.80 |
| | Mean snow coverage in early summer | 0.17 – 0.27 |
| | Mean snow coverage in late summer | 0.00 – 0.04 |
| Snow coverage in upper catchment | Mean snow coverage in spring | 0.81 – 0.90 |
| | Mean snow coverage in early summer | 0.51 – 0.64 |
| | Mean snow coverage in late summer | 0.02 – 0.09 |

**Table C1.** Calibration parameters for the melt and runoff-routing model structures.

| Structure | Parameter | Description | Calibration range | Units |
|---|---|---|---|---|
| $TIM_1$ | $a_{ice}$ | Temperature factor for bare ice | 2.0e-4 - 7.0e-4 | m we $^{\circ}$C$^{-1}$ hr$^{-1}$ |
| | $a_{snow/firn}$ | Temperature factor for snow/firn | 4.0e-7 - 2.0e-4 | m we $^{\circ}$C$^{-1}$ hr$^{-1}$ |
| $TIM_2$ | $a_{ice}$ | Temperature factor for bare ice | 2.0e-4 - 7.0e-4 | m we $^{\circ}$C$^{-1}$ hr$^{-1}$ |
| | $a_{snow/firn}$ | Temperature factor for snow/firn | 4.0e-7 - 2.0e-4 | m we $^{\circ}$C$^{-1}$ hr$^{-1}$ |
| | $b_{ice}$ | Radiation factor for bare ice | 4.0e-7 - 2.0e-6 | m$^3$ we W$^{-1}$ $^{\circ}$C$^{-1}$ hr$^{-1}$ |
| | $b_{snow/firn}$ | Radiation factor for snow/firn | 4.0e-8 - 4.0e-7 | m$^3$ we W$^{-1}$ $^{\circ}$C$^{-1}$ hr$^{-1}$ |
| $TIM_3$ | $a_{ice}$ | Temperature factor for bare ice | 1.5e-4 - 3.0e-4 | m we $^{\circ}$C$^{-1}$ hr$^{-1}$ |
| | $a_{snow/firn}$ | Temperature factor for snow/firn | 6.0e-5 - 2.0e-4 | m we $^{\circ}$C$^{-1}$ hr$^{-1}$ |
| | $b_{ice}$ | Radiation factor for bare ice | 1.0e-5 - 8.0e-5 | m$^3$ we W$^{-1}$ hr$^{-1}$ |
| | $b_{snow/firn}$ | Radiation factor for snow/firn | 2.0e-7 - 4.0e-6 | m$^3$ we W$^{-1}$ hr$^{-1}$ |
| | $p_2$ | Dynamic snow albedo parameter for Brock et al. (2000) model | 0.01 - 0.4 | |
| $ROR_1$ | $k$ | Mean residence time of reservoir | 1 - 30 | hr |
| | $n$ | Number of reservoirs | 1 - 5 | |
| $ROR_2$ | $k_{ice/soil}$ | Mean residence time of runoff from ice and soil | 0.1 - 5 | hr |
| | $k_{snow/firn}$ | Mean residence time of runoff from snow and firn | 20 - 100 | hr |
| | $n_{ice/soil}$ | Number of reservoirs in ice/soil cascade | 1 - 5 | |
| | $n_{snow/firn}$ | Number of reservoirs in snow/firn cascade | 1 - 5 | |

## Appendix C: GHM calibration parameters

Table C1 lists all of the calibration parameters and their pre-defined calibration ranges for the melt and runoff-routing model structures used during the GHM calibration procedure. The three melt model structures include the classic temperature index model ($TIM_1$), the enhanced temperature-index model proposed by Hock (1999) ($TIM_2$) and the enhanced temperature-index model proposed by Pellicciotti et al. (2005) ($TIM_3$). The two runoff-routing model structures include the single linear reservoir cascade ($ROR_1$) and two linear reservoir cascades in parallel ($ROR_2$).

## Appendix D: Decadal changes in effect size for river discharge signatures

*Author contributions.* JDM undertook all practical elements of this study including regional climate projection downscaling, GHM calibration, 21st century projections, ANOVA, analysis of results and production of figures. He also led the writing of this manuscript. JE and ARB managed the design, commissioning and operation of the hydro-meteorological monitoring used in this study. All co-authors contributed to formulation and discussion of methods used as well as writing of the manuscript.

*Competing interests.* The authors declare they have no competing interests.

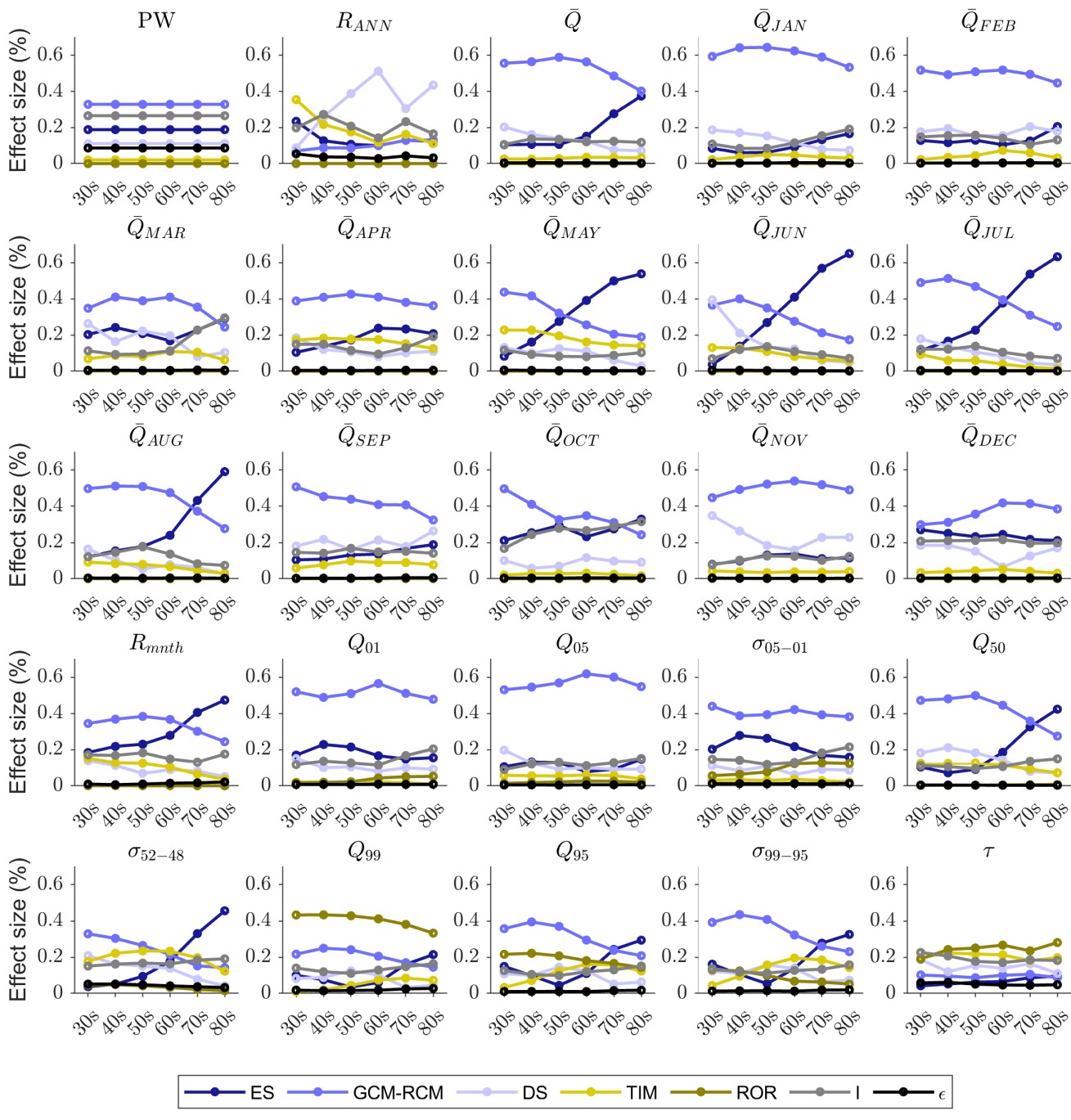

**Figure D1.** Effect size of all main effects, interactions and remaining error on projected decadal changes in the 25 river discharge signatures for all future time-slices centred on the 2030s to the 2080s.

*Acknowledgements.* This work was supported by a NERC studentship awarded to JDM via the Central England NERC Training Alliance (CENTA).

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
