# Peer review of "Future evolution and uncertainty of river flow regime change in a deglaciating river basin"

_Hydrology and Earth System Sciences, 2018_

## Referee Comment (RC1) · Anonymous Referee #1 · 11 Nov 2018

Dear editor, dear authors,

The authors present a climate impact study including uncertainties from different sources of uncertainties. They included an ensemble of RCPs and climate models as well as different model structures into the model chain. Additionally they used many objectives for calibration and for impact assessment (even if the terminus "multi-objective" or "many-objective" was never used in the presented paper). Although the methods of the study need to be more elaborated, the paper presents a comprehensive work. On larger spatial scales multi-impact-model applications have become very common. The authors try to implement such an approach by choosing different, commonly applied

process presentations for the snow/ice processes and for runoff generation processes. For example, runoff generation is represented by two different linear storage methods. However, one limitation with regard to the choice of process representations is that they are, like in the mentioned example, quite similar looking at the variety of process representations possible. In general, the paper is quite long and has a lot of figures. The authors should try to limit word count and balance the length of the different chapters (the discussion is too short compared to other chapters).

I have four main points for the authors to consider.

1. Shortcomings in stepwise calibration The authors calibrate their model in two steps. First, they used different model structures and objectives for calibrating ice/snow processes (TIM) and second they applied different model structures and objectives for calibrating runoff generation (ROC). By calibrating in a first step only different TIM structures, the authors assume that the runoff generation processes will not interact with the TIM processes, which might be true. But I see a problem in the second step of calibration where they use only one TIM representation to further calibrate their model for runoff generation signitures. By this, the calibration is relying only on one set of TIM parameters and its water flux characteristics. I doubt that the selected parameter sets will represent the best possible parameter sets for runoff generation processes. The assumption that the selected 24 different TIM representations will not interact with a particular ROC is in my eyes not correct. I suggest to do a ROC calibration based on all unique TIM representations.

2. Longer periods for impact assessment In climate impact studies it is recommended to take at least 30 years of data for the reference period and a 30 years long future period for analysis. The authors take only 25 years. This limitation should be mentioned in the discussion of methods. Later on they even present their results on climate change impacts on the basis of 10 years slices. This is not acceptable. It is likely that e.g. the results on different uncertainty sources is biased and that the uncertainty coming from the "climate" is overestimated.

3. Separating impacts for different RCPs The authors present quite often a mean over both selected RCPs in the result section and partly in the discussion section. I suggest to show the results for both RCPs separately (or only one) as the results strongly depend on the warming level.

4. Extend discussion The authors discuss the consequences of their findings e.g. with regard to threats for infrastructure and changes in water availability. What I miss is the link of their findings to other studies and actual debates in science. The study would also benefit from a critical discussion of advantages (novelty) and limitations of applied methods.

Details:

Page 2, Line 21: Is "GHM" a common abbreviation for this type of models? This abbreviation is often used for "Global Hydrological Model"

Page 2, Line 30: Upper Yellow is written with capital letters, but upper Indus not

Page 2, Line 31: put "river" before the citation

Page 2, Line 31-33: Why have you decided to include another reference for Q90 that for Q10? Some of the previous mentioned studies (e.g. Vetter 2015) have also analyzed Q90.

Page 3, line 1: ":" could be replaced with "which are"

Page 3, line 27: change "21st century river flows" to "21st century river flow projections"

Page 3, line 28: instead of "emission scenario (ES)", I suggest to use "Representative Concentration Pathway (RCP) "

Page 3, line 35: instead of "large basin-scale" use "regional-scale"

Page 4, line 3-5: I don't think, that all of the hydrological models used in the mentioned studies match the terminology "Computational glacio-hydrological models". I suggest

to use "impact model" instead.

Page 4, line 15: remove "future"

Page 4, line 31: Isn't it 1988-2011 as mentioned in table A1?

Page 4, line 4: Can you include altitude and/or station name of the "terminus"?

Page 5, line 1-2: instead of "than to" use "compared". Where does this numbers come from?

Page 5, line 8: give the installation time for AWS1!

Page 5, line 5-13: Please indicate all the climatic data which was used in your study. I think AWS1 was installed between 2009 and 2011. So there must be additional climate data. If only AWS1 was used, please remove AWS3 and AWS4 from Fig.1.

Why are the uncertainties coming e.g. from precipitation measurements/bias correction/downscaling not included in your study, as the authors did for other objectives (like runoff, snow coverage)?

Page 6, line 14: Replace "to 2100" by "until 2100".

Page 6, line 16: GCM means General Circulation Model and not Global Circulation Model.

Page 6, line 27: Change "the RCP2.6 ES" to "RCP 2.6"

Page 6, line 21-30: I recommend to make a table where all GCM-RCM combinations are shown which were used in this study. This table could be put to the appendix.

Fig.1: The 0.11° EURO-Cordex grid could be indicated in the figure. Page 6, line 31-32: It is common to use at least 30 years, when the climate between two different data sets (here measured and simulated) is compared. Otherwise climate variability is becoming more dominant. If possible, please use 30 years, otherwise this limitation should be clearly mentioned.

Page 6, line 31 - Page 7, line 6: The authors try to analyze the skills of the RCMs by comparing them to the observed climate, but they do it in a very subjective (visual) way, without any statistics or a clear selection criteria. The authors state "Overall, the GCM-RCM performance is good . . ." without any proof or definition, what "good" means. The Authors should explain, why they have not defined any levels of acceptability, as they did for the other inputs.

Fig.2: It is hard to read this figure, especially regarding precipitation. The lines are too close. I guess this is because the extreme precipitation events from climate models are also included. Maybe the 99% quantile is sufficient? Also the ECDF curve for precipitation should start somewhere above the X-axes cross section, indicating the number of days without precipitation.

Fig.3: Instead of writing " based on the projections from 1 of the 14 GCM-RCMs with the RCP8.5 ES" you can give the exact name of the GCM/RCP combination which you refer to.

Page 7, line 19-20: It is common in impact studies to use at least 30 years for reference period and scenario period, respectively.

Page 7, line 25-28: Why don't you calculate deltas as 25-years (or 30-years) moving window? Then you would not need any interpolation.

Page 9, line 5-8: You could put the information about the number of sub-samplings directly to step 5. There are two possible (re)sampling schemes: with or without re-placement. Why have you chosen sampling without replacement?

Page 9, line 12: What is the resolution of you model domain?

Page 10, line 3: Are these new methods to calculate soil infiltration and evapotranspiration? Can you name the methods used to calculate infiltration and evapotranspiration? I can't find/access the given reference Griffiths et al., (2006) in the internet.

Page 10, line 5: given reference Ponce (2014) is gray literature. Chapter is also not

given.

Page 10, line 11: Please be more precise here. What kind of observed data was used? How long are the observational periods.

Page 10, line 31: "The majority of signatures were selected from past studies ..." please give a reference for this statement.

Page 12, line 5: Data description used to create 12 snow and ice signatures is missing

Page 12, line 28: Why always 5000 runs? E.g. for TIM1 you have 2 parameters, but for TIM3 5 parameters. Do you think that this makes a difference?

Page 13, line 5:"Accordingly, only those TIM compositions that captured this signature within the LOA were considered and the rest were discarded." I have the feeling that this is a very strong assumption. What would be the consequence of having a perfect/better DEM and as a result reducing observational uncertainties? You would probably need to discard most of your simulations.

Page 13, line 10: "... best captures the 11 remaining ..." should be 10 instead

Page 14: The whole chapter 2.5.3 is about results and not about methods.

Page 14, line 23: "use model runs" instead of "model chain runs"

Page 14 line 27- Page 16 line 15: This paragraph should be shortened and transferred to the section on model description.

Page 16, line 19-20: It is common to take at least time slices of 30 years for climate impact assessment. By using only 10 years, you will increase the influence of climate variability" and consequently you are overestimating ES uncertainty. I also ask myself whether you have put the duplicated TIM and ROR parameterizations into the ANOVA? How many unique parameterizations do you have for a given TIM structure and a given ROR structure, respectively?

Page 16, line 23: Please check your calculation. I get 1x10ˆ8 (102.850.020)

Page 18, line 3: I suggest to write "[...] consistently show an increase relative to the reference period [. . .]" instead of "consistently predict an increase relative to the recent past"

Page 18, line 4: "The largest increases are predicted [. . .]" Please always use project and not predict for climate projections.

Page 18, line 7-8: Please present climate impacts separate for each RCP. Averaging over both RCPs is not very common.

Page 18, line 14: correct "Figure 5l"

Page 18, line 15: change "The sign of change" into "The direction of change"

Page 18, line 28: Maybe I missed that, but what exactly is "annual snow coverage"?

Fig.7: Why are the snow coverage confidence bands only positive compared to the mean? Does snow coverage also include ice, or does it only count as snow coverage if there is snow on the glacier?

Page 20, line 12: improve English: "The maximum coverage simulations show higher than . . ."

Fig 5. instead of "recent past " use reference period

Page 20, line1: "projects" instead of "predicts"

Line 20, line 1-10: Again: I guess these results are highly dependent on the warming levels and it would be good to either choose one RCP or to analyze both separately.

Fig. 11: Could you please present the results in mm and not in km$^3$, because mm is more intuitive and easier to compare to other studies. Is snow included in rainfall? In previous plots you named it always "total precipitation".

[Figure]

---

## Referee Comment (RC2) · Zappa (Referee) · 5 Dec 2018

**Future evolution and uncertainty of river flow regime change in a deglaciating river basin**

Jonathan D Mackay1,2, Nicholas E Barrand1, David M Hannah1, Stefan Krause1, Christopher R Jackson2, Jez Everest3, Guðfinna Aðalgeirsdóttir4, and Andrew R Black5

[referee-annotated manuscript omitted]

---

## Author Comment (AC1) · 20 Dec 2018

**hess-2018-443 response to anonymous referee #1 20/12/2018**.

We thank anonymous referee #1 (R1) for taking their time to read our manuscript and provide us with a very thorough and fair set of comments and recommendations. We are largely in agreement with the recommendations and have therefore proposed some substantial revisions that we feel will result in a much-improved manuscript. We have considered our response to each comment carefully, especially to those that require clarification or are critical for which we provide more detailed responses with clear justifications. We have listed all referee comments in black followed by our responses in blue. Any proposed revisions are highlighted with **bold** text. We did notice that R1 included incorrect page numbers on several occasions. For the reader's convenience, we have corrected these in this response using orange text. We look forward to the response from the editor at their earliest convenience.

Dear editor, dear authors,

The authors present a climate impact study including uncertainties from different sources of uncertainties. They included an ensemble of RCPs and climate models as well as different model structures into the model chain. Additionally they used many objectives for calibration and for impact assessment (even if the terminus "multi-objective" or "many-objective" was never used in the presented paper). Although the methods of the study need to be more elaborated, the paper presents a comprehensive work. On larger spatial scales multi-impact-model applications have become very common. The authors try to implement such an approach by choosing different, commonly applied process presentations for the snow/ice processes and for runoff generation processes. For example, runoff generation is represented by two different linear storage methods. However, one limitation with regard to the choice of process representations is that they are, like in the mentioned example, quite similar looking at the variety of process representations possible. In general, the paper is quite long and has a lot of figures. The authors should try to limit word count and balance the length of the different chapters (the discussion is too short compared to other chapters).

There are two points here which are not explicitly raised the remainder of the review and so we will address these before proceeding to the detailed comments.

> i) *"In general, the paper is quite long and has a lot of figures. The authors should try to limit word count and balance the length of the different chapters (the discussion is too short compared to other chapters)."*

As R1 appreciates, this manuscript *"presents a comprehensive"* piece of work and accordingly we took great care in choosing the level of detail to ensure reproducibility and to achieve the two principal aims of the study outlined in the introduction. While 14 figures and two tables in the main text is perhaps more than average, we don't feel it is excessive, particularly for a study as comprehensive as this. Indeed, each figure has a clear purpose and demonstrates something new to the reader that cannot be deduced from any of the other figures or from the text. For similar reasons, we don't feel the word count is excessive. Note, nearly 60% of the wording is contained within the methods and discussion sections and we are in agreement with R1 that there is scope to elaborate further on some aspects of the methodology and to extend the discussion. Even so, **on revising the manuscript, we will make every attempt to limit the word count where possible.**

> ii) *"…one limitation with regard to the choice of process representations is that they are, like in the mentioned example, quite similar looking at the variety of process representations possible."*

We appreciate that there are many other runoff-routing models that have been used in glaciated catchments ranging from linear reservoir models to physically-based hydraulic models that simulate discrete flow pathways through the glacier (e.g. Arnold et al. 1998). We adopted the concept of linear reservoirs because they have been so widely used in other glacier-hydrology studies (Hock and Jansson, 2005), particularly in climate change impact assessments. Their popularity can be partly explained by their simplicity. They employ few parameters, making them ideal for data-scarce mountain catchments. Additionally, the concept of linear reservoirs lends itself to structural modifications (e.g. through implementing multiple reservoirs in series or parallel) making them extremely versatile across different glaciated settings. Therefore, an investigation into the uncertainty stemming from using linear-reservoirs is particularly relevant to the field of glaciated catchment hydrology. As we state in the manuscript (P10L7), we selected the two models implemented in this study based on results from our previous study (Mackay *et al.*, 2018) where we experimented with three different levels of linear-reservoir complexity. The most complex structure which implemented separate reservoirs for snow, firn, ice and soil actually inhibited model efficiency and therefore we could not justify including it in this study. Indeed, the fact that relatively simple (and more similar) structures were best-suited for this particular catchment is perhaps not surprising. As we note (P30L14), "previous investigations have shown that Virkisjökull has a well developed conduit drainage system that routes runoff efficiently year-round (Phillips et al., 2014; Flett et al., 2017)." This is contrary to many other temperate glaciers which exhibit transient water storage behaviour due to the periodic (seasonal) activation of efficient englacial/subglacial drainage pathways (Jansson, Hock and Schneider, 2003). For these catchments, the use of more complex model runoff-routing model structures might also be justifiable. Even so, we appreciate that if a model interrogation study like that of Mackay *et al.* (2018) were undertaken which evaluated other runoff-routing models, it could yield additional 'suitable' model structures to be included in a study like this. This of course is beyond the scope of this study, but is an **important point to raise in the discussion and will be done so in the revised manuscript**.

I have four main points for the authors to consider.

1. Shortcomings in stepwise calibration: The authors calibrate their model in two steps. First, they used different model structures and objectives for calibrating ice/snow processes (TIM) and second they applied different model structures and objectives for calibrating runoff generation (ROC). By calibrating in a first step only different TIM structures, the authors assume that the runoff generation processes will not interact with the TIM processes, which might be true. But I see a problem in the second step of calibration where they use only one TIM representation to further calibrate their model for runoff generation signatures. By this, the calibration is relying only on one set of TIM parameters and its water flux characteristics. I doubt that the selected parameter sets will represent the best possible parameter sets for runoff generation processes. The assumption that the selected 24 different TIM representations will not interact with a particular ROC is in my eyes not correct. I suggest to do a ROC calibration based on all unique TIM representations.

To be clear, the assertion that *"only one TIM representation [is used] to further calibrate their model for runoff generation signatures"* is not correct. As we state (P13L25), "a single [ROR] parameter set was selected based on its mean performance across the 24 TIM compositions". In other words, we used all 24 TIM parameter-structure combinations (compositions) in combination with all potential ROR compositions (5000 parameter sets × 2 structures) and selected the most efficient ROR models based on their overall (mean) performance across the 24 TIM compositions. The reason we selected the ROR models in this way was because the second aim of the manuscript necessitated it (P4L18). Namely, "to use ANOVA to quantify the relative influence of the five model chain components to

projection uncertainty across the different characteristics of river flow regime". More specifically, in order to apply ANOVA and separate out the uncertainties from the TIM and ROR model chain components (factors), we needed an ensemble of model chain runs that included every combination of each factor level so that equations 2-7 could be solved. If we had selected the best ROR compositions separately for each of the 24 TIM compositions, we would not have been able to do this. We emphasise this point in the manuscript (P13L26) by stating that this ROR model selection process "…was done to satisfy the ANOVA requirements so that the TIM and ROR composition uncertainty could be analysed separately".

As R1 rightly points out, a limitation of this stepwise calibration approach is that it does not fully take into account interactions between the TIM and ROR model components given that they are not calibrated simultaneously. Indeed, we sympathise with R1's doubts that because of this *"the selected parameter sets will [not] represent the best possible parameter sets for runoff generation processes"*. We would however, emphasise that the model evaluation (Figure 4) demonstrates that the models are able to capture the majority of the signatures within their observation uncertainty bounds. Furthermore, the ANOVA assessment indicates that for future projections at least, interactions between the TIM and ROR models are negligible except for two of the 25 signatures evaluated (Figure 14). Even so, we agree that it is important to emphasise the assumptions made by adopting this calibration approach and **we will this highlight this as a potential limitation in the discussion section of the revised manuscript.**

2. Longer periods for impact assessment: In climate impact studies it is recommended to take at least 30 years of data for the reference period and a 30 years long future period for analysis. The authors take only 25 years. This limitation should be mentioned in the discussion of methods. Later on they even present their results on climate change impacts on the basis of 10 years slices. This is not acceptable. It is likely that e.g. the results on different uncertainty sources is biased and that the uncertainty coming from the "climate" is overestimated.

We certainly agree with R1 that when analysing climate projection time-series for temporal changes in climate, the use of longer time-slices help to smooth out year-to-year climate variability and therefore any derived shifts in their statistical properties are likely to be more robust. Indeed, we can reassure R1 that when defining our reference and future climate periods we made them as long as possible.  25 years was selected because our observed climate data only extends back to 1981 and the EURO-CORDEX historic simulations only run up to the end of 2005. We were therefore limited to this 25-year period to represent our reference climate data period.

We disagree that using 25 years is a significant limitation of the study. Climate impact studies for Iceland have typically used a shorter, 20-year (1981-2000) reference climate period (Aðalgeirsdóttir *et al.*, 2006, 2011; Guðmundsson *et al.*, 2009). The 1981-2000 reference period has also been used to evaluate hydrological impacts in other glaciated regions (e.g. Liu *et al.*, 2013) and in the recently published state-of-the art UK Climate Projections (UKCP18 – see https://www.metoffice.gov.uk/research/collaboration/ukcp) which evaluate changes in means and quantiles of rainfall and temperature like our study.

We sympathise with R1's reservations about using 10-year time-slices to evaluate changes and uncertainties in the hydrological projections. We could have used 25-year time-slices to maintain consistency with the climate data and to bolster the robustness of any derived changes and uncertainties. We chose 10-year time-slices to provide a more refined indication of changes in the signatures and their sources of uncertainty over time, but on reflection, we agree that by doing so we may have introduced biases in inter-annual variability. **We will therefore re-analyse the results**

**using a new set of 25-year long time-slices in accordance with the driving climate data. For this, 1991-2015 will be used as the reference period** (note, we do not have ice-coverage data prior to 1988 and so can't use the 1981-2005 period used for the climate) **and a set of future periods centred on the 2030s (2023-2047), 2040s (2033-2057), 2050s (2043-2067), 2060s (2053-2077), 2070s (2063-2087) and 2080s (2073-2097) will be used.**

3. Separating impacts for different RCPs: The authors present quite often a mean over both selected RCPs in the result section and partly in the discussion section. I suggest to show the results for both RCPs separately (or only one) as the results strongly depend on the warming level.

The decision to combine the RCPs was made for two reasons. Firstly, as the referee has noted, the manuscript outlines a comprehensive piece of work and we were hesitant to include the additional text and/or figures to present results from the different RCPs separately. The majority of hydrological impact studies that use multiple emission scenarios display their respective projections separately as these are often the main/only source of uncertainty investigated. In our study, we have included five different sources of uncertainty and it is clear that the emission scenario dominates snow and ice projection uncertainty (Figure 10). **We therefore agree that it would be beneficial to separate out the projections in Figure 7 by RCP and update the text in the results section accordingly in the revised manuscript.** For the hydrological projections, the emission scenario plays an important (although rarely dominant) role in projection uncertainty (see Figures 13 and C1). The decision to prioritise the separation of the RCPs over say the GCM-RCMs for the hydrological projections (which dominate projection uncertainty more frequently) is therefore questionable. However, it would allow for much easier comparison with previous studies, and so **we will separate the RCPs for Figures 11 and 12 also in the revised manuscript.**

4. Extend discussion: The authors discuss the consequences of their findings e.g. with regard to threats for infrastructure and changes in water availability. What I miss is the link of their findings to other studies and actual debates in science. The study would also benefit from a critical discussion of advantages (novelty) and limitations of applied methods.

We appreciate that section 4.1 of the discussion is weighted towards impact assessment although we do feel relating the projections to downstream impacts forms and important part of any climate change impact experiment. We agree that there is scope to expand certain aspects of the discussion. In particular, in the revised manuscript **we will discuss how our results compare to those of other glacio-hydrological uncertainty analyses (as outlined P3L20-30). We will also make it clearer to the reader what the novelties of our approach are (as outlined P4L7-12) and exactly what they have shown us**. Finally, **we will include additional text to discuss potential limitations of our approach.** Several of these limitations have already been outlined in this response and several more are detailed in the text that follows.

Page 2, Line 21: Is "GHM" a common abbreviation for this type of models? This abbreviation is often used for "Global Hydrological Model"

We decided to abbreviate glacio-hydrological model for this manuscript because we use the term 55 times. While we appreciate that GHM is also used to abbreviate Global Hydrological Model, we don't see that as being particularly confusing for a reader of this manuscript given the focus is on a glaciated catchment-based study and the only mention of global-scale hydrological modelling is written as such (P4L1). We also make it clear to the reader at the beginning of the introduction (P2L21) that we are using GHM to refer to glacio-hydrological model. For these reasons we would like to keep the GHM abbreviation in the revised manuscript.

Page 2, Line 30: Upper Yellow is written with capital letters, but upper Indus not

**This will be corrected in the revised manuscript.**

Page 2, Line 31: put "river" before the citation

**This will be corrected in the revised manuscript.**

Page 2, Line 31-33: Why have you decided to include another reference for Q90 that for Q10? Some of the previous mentioned studies (e.g. Vetter 2015) have also analyzed Q90.

The choice of citations for high and low flow projections in glaciated basins was made based on those that robustly showed changes in these variables (i.e. with confidence). On re-reading this paragraph, we appreciate that we did not make this clear and instead it reads as though we're saying only those studies have attempted to project changes in high/low flows. Accordingly, **we will re-write this to make this clear**.

We also feel it is important to highlight that projections in low flow magnitudes for glaciated river basins have in general been less conclusive, **and we will include the Vetter study as an example of this**.

Finally, we noted that we could have included the Wijngaard *et al.* (2017) study, which clearly shows shifts in low flow quantiles, and **as such, we will include this in the revised manuscript**.

Page 3, line 1: ":" could be replaced with "which are"

**Agreed. This will be corrected in the revised manuscript.**

Page 3, line 27: change "21st century river flows" to "21st century river flow projections"

**Agreed. This will be corrected in the revised manuscript.**

Page 3, line 28: instead of "emission scenario (ES)", I suggest to use "Representative Concentration Pathway (RCP) "

We use the term emission scenario (ES) to broadly refer to all types of emission scenarios of which RCPs are one type. For this reason, we decided not to adopt the term RCP as the Jobst *et al.* (2018) study uses the older SRES emission scenarios. We also clarify that the ESs we use in this study are of type RCP (P6L23).

Page 3, line 35: instead of "large basin-scale" use "regional-scale"

**Agreed. This will be corrected in the revised manuscript.**

Page 4, line 3-5: I don't think, that all of the hydrological models used in the mentioned studies match the terminology "Computational glacio-hydrological models". I suggest to use "impact model" instead.

Yes, we agree that the hydrological models used in these studies are not strictly glacio-hydrological models. The reason we used this term was to thread together glacio-hydrological modelling and uncertainty analysis in the introduction but we appreciate that by doing so this statement is somewhat misleading. Accordingly, **we will change the term "GHM" (which is used four times in this paragraph) to "hydrological model" in the revised manuscript.**

Page 4, line 15: remove "future"

**This will be corrected in the revised manuscript.**

Page 4, line 31: Isn't it 1988-2011 as mentioned in table A1?

Yes it is. **This will be corrected in the revised manuscript.**

Page 5, line 4: Can you include altitude and/or station name of the "terminus"?

**Yes, we will add the AWS1 station name to the revised manuscript.**

Page 6, line 1-2: instead of "than to" use "compared". Where does this numbers come from?

Agreed, **"than to" will be changed to "compared" in the revised manuscript.**

The numbers come from the gridded precipitation product detailed in section 2.1.1 (P6L11). **We will include the Nawri *et al.* (2017) reference in the revised manuscript to make this clear.**

Page 6, line 8: give the installation time for AWS1!

AWS1 and AWS4 were installed in 2009 and 2011 respectively and **this will be included in the revised manuscript.**

Page 6, line 5-13: Please indicate all the climatic data which was used in your study. I think AWS1 was installed between 2009 and 2011. So there must be additional climate data.

Yes, we appreciate we have been too brief on the climate data used and simply refer the reader to the Mackay et al. (2018) study for further information. **Accordingly, we will detail all of the climate data in the revised manuscript**.

If only AWS1 was used, please remove AWS3 and AWS4 from Fig.1.

No, both AWS1 and AWS4 were used as will be detailed in the revised manuscript. **We will remove AWS3 from Figure 1 in the revised manuscript.**

Why are the uncertainties coming e.g. from precipitation measurements/bias correction/downscaling not included in your study, as the authors did for other objectives (like runoff, snow coverage)?

This is an excellent question that we did not address adequately in the discussion, but one we have thought about greatly. Undoubtedly, uncertainties in the driving observation precipitation data, whether from observation error (e.g. at the gauge) or post-processing errors (e.g. bias-correction of the gridded precipitation data) could contribute significantly to projection uncertainty. Indeed, we thought carefully about incorporating these uncertainties into our experiment, but decided against it for two reasons:

1) It would have required us to define a statistical model of the precipitation uncertainties. As noted by R1, we did this for other variables including river discharge, for which we used a method which has been applied in a variety of river basin settings around the world (McMillan and Westerberg, 2015). This was possible because we had the information and data required to compute these uncertainties. Models of precipitation uncertainty have been used in the past e.g. see Blazkova and Beven (2009) who use a precipitation uncertainty model in a hydrological model selection approach. However, the structure and parameterisation of such a model would have been difficult to define in our study, given that we have almost no information on rain gauge errors. We also have very little information on biases in the gridded ICRA precipitation dataset (particularly at high elevations where the

gauge network is sparse). Accordingly, we felt that any attempt to incorporate these uncertainties into our study would be difficult to justify and could potentially introduce additional biases that would be detrimental to the robustness of the projections.

2) Of course, 1) would not have precluded us from evaluating uncertainties in the quantile-mapping approach used to bias-correct the ICRA dataset. For example, one could imagine undertaking some type of bootstrapping experiment to evaluate the uncertainties in these models. However, this would have required us to include an additional (sixth) factor in our ANOVA experiments. By doing this, we would have made our ensemble size n-times larger, where n is the number of precipitation time-series. With 94,080 simulations, we were already at very limit of what was computationally feasible (even running on an HPC) and therefore we decided against introducing this as an additional source of uncertainty.

Nevertheless, the fact that we are not accounting for potentially very high uncertainties in the historical precipitation data is a limitation of this work and, as such, **we will include this as an additional discussion point in the revised manuscript.**

Page 6, line 14: Replace "to 2100" by "until 2100".

**This will be corrected in the revised manuscript.**

Page 6, line 16: GCM means General Circulation Model and not Global Circulation Model.

**This will be corrected in the revised manuscript.**

Page 6, line 27: Change "the RCP2.6 ES" to "RCP 2.6"

Yes, **ES will be removed**, but note, RCP and 2.6 should not have a space between them and so this will not been modified as suggested.

Page 6, line 21-30: I recommend to make a table where all GCM-RCM combinations are shown which were used in this study. This table could be put to the appendix.

Yes, we agree this would be a useful addition to the manuscript. **We will include this in the revised manuscript.**

Fig.1: The 0.11◦ EURO-Cordex grid could be indicated in the figure.

**Yes we're happy to add the CORDEX grids to the manuscript (note not all RCMs use the same grid) and will do so in the revised manuscript.**

Page 6, line 31-32: It is common to use at least 30 years, when the climate between two different data sets (here measured and simulated) is compared. Otherwise climate variability is becoming more dominant. If possible, please use 30 years, otherwise this limitation should be clearly mentioned.

We've addressed this in the above comments.

Page 6, line 31 - Page 7, line 6: The authors try to analyze the skills of the RCMs by comparing them to the observed climate, but they do it in a very subjective (visual) way, without any statistics or a clear selection criteria. The authors state "Overall, the GCMRCM performance is good . . ." without any proof or definition, what "good" means. The Authors should explain, why they have not defined any levels of acceptability, as they did for the other inputs.

Yes, we appreciate that on trying to keep the word count down, we provided very little information on the climate model skill, and we agree that Figure 2 is not entirely informative of the potential weaknesses of the different GCM-RCMs. In fact, it is important to state that the purpose of including the climate model skill in the manuscript was not to present any kind of formal climate model selection strategy. Rather, we were trying to be transparent about the potential weaknesses/biases in the GCM-RCMs that were used to drive the projections. On reflection, a table of statistics would have been much more informative. Accordingly, **for the revised manuscript we will include a table of statistics for each GCM-RCM showing seasonal average means of each climate variable as well as lower and upper percentiles (e.g. 1 and 99 – just 99 for precipitation) to indicate these biases.**

We avoided employing a limits of acceptability approach given the highly non-linear nature of climate model projections and the difficulty in evaluating historic skill and relating it to future projection skill. We did however decide to remove one of the GCM-RCMs from the analysis entirely given it's relatively poor fit to observation data, particularly for the temperature data. More specifically, we found on comparing monthly average temperatures (Figure R1a below) that the [CNRM-CM5]-[ALADIN53] GCM-RCM shows anonymously large negative biases, particularly during the winter months of the year. A comparison of the ECDFs for the winter months also shows these biases (Figure R1b, c and d). We also calculated the RMSE of the monthly ECDFs for each GCM-RCM and as we stated in the manuscript the [CNRM-CM5]-[ALADIN53] "was consistently poor across all three climate variables". More specifically, when ranked according to their RMSE scores, the [CNRM-CM5]-[ALADIN53] GCM-RCM ranked 14, 13 and 15 out of 15.

Given the anonymously high biases in temperature and the importance of temperature for driving hydrological change in the catchment (both in terms of melt rate and the proportion of precipitation falling as rainfall), coupled with the fact that the model was relatively poor across all three climate variables, it was deemed appropriate to remove this model from the ensemble. **In the revised manuscript we will include additional text to explain our decision in removing this GCM-RCM from the ensemble.**

[Figure]

Figure R1: Comparison of observed and simulated (GCM-RCMs) monthly mean temperatures for the recent past (1981-2005)

Fig.2: It is hard to read this figure, especially regarding precipitation. The lines are too close. I guess this is because the extreme precipitation events from climate models are also included. Maybe the 99% quantile is sufficient? Also the ECDF curve for precipitation should start somewhere above the X-axes cross section, indicating the number of days without precipitation.

Yes, we have proposed using a table of statistics to replace Figure 2.

Fig.3: Instead of writing " based on the projections from 1 of the 14 GCM-RCMs with the RCP8.5 ES" you can give the exact name of the GCM/RCP combination which you refer to.

**Agreed. This will be corrected in the revised manuscript.**

Page 7, line 19-20: It is common in impact studies to use at least 30 years for reference period and scenario period, respectively.

We've addressed this in the comments above.

Page 7, line 25-28: Why don't you calculate deltas as 25-years (or 30-years) moving window? Then you would not need any interpolation.

The reason we settled on using a linear model for perturbing the deltas over time was because, to our knowledge at least, the study of Farinotti *et al.* (2012) is the only study that directly addresses the need for transient deltas for the purpose of dynamic glacier evolution modelling for hydrological impact assessment. Given the successful application in this study to a number of catchments of similar size and setting (Swiss Alps), we decided it would be most appropriate and justifiable to adopt this approach also. We recognise though, that we could have chosen a different approach (such as a moving average window). This is therefore a potential source of uncertainty that we have not accounted for in our projections and **in our revised manuscript we will include this as an additional limitation in the discussion**.

Page 9, line 5-8: You could put the information about the number of sub-samplings directly to step 5. There are two possible (re)sampling schemes: with or without replacement. Why have you chosen sampling without replacement?

On reflection we agree that this would sit better as part of step five and we **will edit the manuscript accordingly as part of the revisions.**

To be clear, we are sampling with replacement not without replacement as suggested. **We will make this clear in step five of the revised manuscript**.

Page 9, line 12: What is the resolution of you model domain?

It's 50 m. **We will add this to the revised manuscript**.

Page 10, line 3: Are these new methods to calculate soil infiltration and evapotranspiration? Can you name the methods used to calculate infiltration and evapotranspiration? I can't find/access the given reference Griffiths et al., (2006) in the internet.

No, they are not new methods. The soil moisture model is a sub-model of the UK Centre for Ecology and Hydrology's Continuous Estimation of River Flows (CERF) regional hydrological model. It's based on the well-established FAO56 soil moisture accounting procedure (Allen *et al.*, 1998) and has shown to compare favourably to physically-based models at the field scale where interception losses are small (Sorensen *et al.*, 2014). It has been used in the past for national scale hydrological river and groundwater level projections under climate change (Prudhomme *et al.*, 2013; Collet *et al.*, 2018), catchment scale groundwater level modelling (Mackay, Jackson and Wang, 2014; Mackay *et al.*, 2015; Jackson *et al.*, 2016) and distributed recharge modelling (Mansour *et al.*, 2018). We appreciate that the reader would benefit from knowing that the model has been used extensively in the past and so **we will include some of the above citations in the revised manuscript to indicate this.**

It appears that the document we cited is no longer freely available to access online, but an identical document (albeit published by the Environment Agency for England and Wales two years later) can

be found here: https://www.gov.uk/government/publications/continuous-estimation-of-river-flows. **We will update this reference in the revised manuscript**.

Page 10, line 5: Given reference Ponce (2014) is gray literature. Chapter is also not given.

We chose to reference the most recent edition of this book which can only be accessed online. However, we appreciate R1's preference for formally published literature and on reading the first edition published by Prentice-Hall in 1989, the section on linear reservoir models is identical. **Accordingly, we will reference the published first edition copy in the revised manuscript with page numbers.**

Page 10, line 11: Please be more precise here. What kind of observed data was used? How long are the observational periods.

This information is given in appendix A and referred to in the subsequent "GHM calibration" section. However, we appreciate that the reader may wish to know more about these data on the first mention of "observation data of ice melt and snow coverage". Accordingly, **in the revised manuscript we will included a reference to this appendix at the end of this sentence.**

In addition, **we also intend to add some extra information to appendix A including the months used to define the seasonal snow coverage signatures, the elevation ranges of the lower, middle and upper sections of the glacier-free basin area and the years for which the MOD10A1 MODIS data cover.**

Page 10, line 31: "The majority of signatures were selected from past studies . . ." please give a reference for this statement.

Yes, we did not include these references as they are given in Mackay et al. (2018), **but we will include them in the revised manuscript.**

Page 12, line 5: Data description used to create 12 snow and ice signatures is missing

This can be found in the appendix as referenced in the same sentence (P12L6).

Page 12, line 28: Why always 5000 runs? E.g. for TIM1 you have 2 parameters, but for TIM3 5 parameters. Do you think that this makes a difference?

Undoubtedly, the larger parameter space of $TIM_3$ means the calibrated models could be 'less optimal' than the $TIM_1$ models, which has three fewer calibration parameters. As with any Monte Carlo calibration procedure, we tried to balance the density of parameter sampling with our available computational resources (note while the GHM is relatively efficient, it still requires considerable computation time given its distributed structure and hourly timestep). Indeed, we adopted the Sobol parameter sampling strategy to ensure we sampled the parameter space as evenly as possible given these computational limitations. We would emphasise that the calibrated models did show a good fit to the majority of hydrological signatures (Figure 4) and accordingly we are happy that the calibration procedure worked. However, we recognise, that given more computational resources, we would have undertaken a denser parameter sampling, particularly for the more complex model structures. We appreciate that we have not addressed this in the manuscript **and will therefore include under-sampling of the parameter space as a potential limitation in the discussion of the revised manuscript.**

Page 13, line 5:"Accordingly, only those TIM compositions that captured this signature within the LOA were considered and the rest were discarded." I have the feeling that this is a very strong

assumption. What would be the consequence of having a perfect/better DEM and as a result reducing observational uncertainties? You would probably need to discard most of your simulations.

We agree that if we were to have perfect observations (although of course in reality this is impossible), you would need a perfect model to capture those observations within their uncertainty bounds. Such a result may lead one to obtain no acceptable models leading them to reconsider their model structure and/or parameterisation. They may of course also decide that such LOA are simply impossible to match given limitations/uncertainties associated with boundary conditions/model resolution etc. and in such cases one may choose to relax their LOA (e.g. see Blazkova and Beven, 2009).

We're not entirely sure what the "strong assumption" alluded to in the above comment refers to? Arbitrary acceptability thresholds (e.g. Nash-Sutcliffe > 0.6 for river flows) are routinely used in hydrological studies to discard large numbers of "unacceptable" or "non-behavioural" models. We would deem the assumptions underlying these approaches much more suspect given they provide little information on what characteristics of model behaviour/processes the behavioural models capture and little reasoning as to why an NSE of 0.6 is good enough. We instead have used multiple model evaluation metrics (signatures) which relate to different aspects of model behaviour and, by quantifying their uncertainty we can define an acceptable model as one that captures the observations within their uncertainty bounds. This also allows us to make more robust statements about model acceptability and (crucially) model deficiencies (e.g. Figure 4d).

As we state in the manuscript (P13L3), "Given the high degree of glaciation in the study catchment, and its recent rapid retreat, an initial emphasis of the calibration was put on the model's ability to capture the long term glacier volume change signature."

In other words, given its recent retreat, the glacier has the potential to retreat much further over the 21$^{st}$ century and this could significantly alter the hydrological regime of the catchment (our projection metrics). Given the sensitivity of the projections to the projected glacier retreat (note summer runoff from ice melt was approximately double that of rainfall in the 1990s – Figure 11), we feel we are justified in our decision to use the ice volume change signature to prune the initial set Monte-Carlo runs.

Page 13, line 10: ". . . best captures the 11 remaining . . ." should be 10 instead

Agreed, **we will correct this in the revised manuscript.**

Page 14: The whole chapter 2.5.3 is about results and not about methods.

We agree, this chapter would sit much better in the results section. **We will move this chapter to the beginning of the results section in the revised manuscript.**

Page 14, line 23: "use model runs" instead of "model chain runs"

Agreed. **This will be corrected in the revised manuscript.**

Page 14 line 27- Page 16 line 15: This paragraph should be shortened and transferred to the section on model description.

Agreed. **This will be corrected in the revised manuscript.**

Page 16, line 19-20: It is common to take at least time slices of 30 years for climate impact assessment. By using only 10 years, you will increase the influence of climate variability'' and consequently you are overestimating ES uncertainty.

We've addressed this in the previous responses.

I also ask myself whether you have put the duplicated TIM and ROR parameterizations into the ANOVA?

No, we did not include duplicates into the ANOVA given that we are sub-sampling all factors down to two levels and wanted to avoid generating sums of squares equal to zero. We appreciate we should have stated this in manuscript given that it could influence the results shown. Given that we will need to run the ANOVA analysis again, **we are happy to experiment with including the duplicates in the sub-sampling and assess for convergence of results according to P16L18-19.**

How many unique parameterizations do you have for a given TIM structure and a given ROR structure, respectively?

As we state (P14L2), there are "24 unique TIM compositions were obtained from this calibration stage made up of eight unique parameterisations of each of the three TIM structures." Also, we state (P15L5) that "14 unique ROR compositions were selected made up of seven unique parameterisations of the ROR1 and ROR2 structures, giving a total of 24×14 = 336 unique GHM compositions".

Page 16, line 23: Please check your calculation. I get 1x10^8 (102.850.020)

Yes, in our defence the statement and calculation is correct i.e. there are ~ $4*10^9$ unique combinations of factors levels when sub-sampled down to two. However, a large number of these would include level repetitions for at least one factor and indeed, we did not sub-sample these in our study making this statement inconsistent with the applied methods. **We will amend this in the revised manuscript.**

Page 18, line 3: I suggest to write "[...] consistently show an increase relative to the reference period [. . .]" instead of "consistently predict an increase relative to the recent past

While we appreciate the referee's preference for using "reference period" rather than "recent past", we feel that we have made it clear to the reader exactly what we mean by "recent past" in the methodology section. Also, we use the term "reference period" to refer to the changes in hydrological signatures which must have a different reference period to the climate (see previous responses). Accordingly, to avoid confusion we would prefer to keep the term "recent past". **We do however agree that the word "predict" should be replaced with "show" and this modification will be made in the revised manuscript.**

Page 18, line 4: "The largest increases are predicted [. . .]" Please always use project and not predict for climate projections.

**This will be corrected in the revised manuscript along with the two other occurrences of the word "predict" (P20L1 and P24L4).**

Page 18, line 7-8: Please present climate impacts separate for each RCP. Averaging over both RCPs is not very common.

Agreed. **We'll amend these statements in the revised manuscript.**

Page 18, line 14: correct "Figure 5l"

We're referring figure 5l represents changes in precipitation for the autumn months.

Page 18, line 15: change "The sign of change" into "The direction of change"

**This will be corrected in the revised manuscript along with the eight other identified occurrences of this phrase (P1L11, P22L8, P24L10, P25L3, P25L7, P29L34, P30L21, P31L23).**

Page 18, line 28: Maybe I missed that, but what exactly is "annual snow coverage"?

We mean the "annual mean watershed snow coverage" which we realise we did not make clear to the reader. **We will revise this wording along with the wording in the caption of figure 7 to make this clear.**

In addition, we noted that we express the snow coverage signatures as the proportion of glacier-free basin area covered by snow in Table A1 rather than the total coverage in $km^2$ as in Figure 7. To avoid confusing the reader, **we will add a sentence to the end of the caption for this table in the revised manuscript which reads: "Note, snow coverage is expressed as a proportion of the glacier-free basin area."**

Fig.7: Why are the snow coverage confidence bands only positive compared to the mean? Does snow coverage also include ice, or does it only count as snow coverage if there is snow on the glacier?

We appreciate it looks as though the confidence bounds are always higher than the mean (note the mean is not actually shown in Figure 7), but the confidence bounds do in fact extend below the mean. What you're seeing is the tail-end of the ensemble above the mean. The majority of simulations reside near the lower bound (i.e. the distribution of simulations is skewed).

Page 20, line 12: improve English: "The maximum coverage simulations show higher than . . ."

We're not entirely sure what the issue is here. We state (P20L10) that "Figures 9a, b and c show the climate projection time-series that produced the minimum (dotted lines) and maximum (dashed lines) snow (blue lines) and ice (red lines) coverage by 2100." We then refer to these as the minimum coverage and maximum coverage simulations. We therefore cannot identify a reason to change this phrasing.

Fig 5. instead of "recent past " use reference period

Please see above comment.

Page 20, line1: "projects" instead of "predicts"

Yes, **this will be modified as noted in the previous responses.**

Line 20, line 1-10: Again: I guess these results are highly dependent on the warming levels and it would be good to either choose one RCP or to analyze both separately.

**We will analyse both separately.**

In addition to the proposed revisions above, we also noticed that Figure 4 does not have the a,b,c and d labelling on it. **These labels will be added in the revised manuscript.**

Fig. 11: Could you please present the results in mm and not in km3 , because mm is more intuitive and easier to compare to other studies. Is snow included in rainfall? In previous plots you named it always "total precipitation".

Yes, we can change the units to mm. No rainfall only includes rainfall otherwise we would have named it total precipitation. Note, as described in the figure caption, we're showing the runoff components, hence why we're showing rainfall and not total precipitation here.

**References:**

Aðalgeirsdóttir, G. *et al.* (2006) 'Response of Hofsjökull and southern Vatnajökull, Iceland, to climate change', *Journal of Geophysical Research*, 111(F3), p. F03001. doi: 10.1029/2005JF000388.

Aðalgeirsdóttir, G. *et al.* (2011) 'Modelling the 20th and 21st century evolution of Hoffellsjökull glacier, SE-Vatnajökull, Iceland', *The Cryosphere*, 5(4), pp. 961–975. doi: 10.5194/tc-5-961-2011.

Allen, R. G. *et al.* (1998) *Crop evapotranspiration - Guidelines for computing crop water requirements - FAO Irrigation and drainage paper 56*. Rome, Italy.

Blazkova, S. and Beven, K. (2009) 'A limits of acceptability approach to model evaluation and uncertainty estimation in flood frequency estimation by continuous simulation: Skalka catchment, Czech Republic', *Water Resources Research*, 45(12), p. W00B16. doi: 10.1029/2007WR006726.

Collet, L. *et al.* (2018) 'Future hot-spots for hydro-hazards in Great Britain : a probabilistic assessment', *Hydrology and Earth System Sciences*, (June), pp. 1–22. doi: 10.5194/hess-2018-274.

Farinotti, D. *et al.* (2012) 'Runoff evolution in the Swiss Alps: projections for selected high-alpine catchments based on ENSEMBLES scenarios', *Hydrological Processes*, 26(13), pp. 1909–1924. doi: 10.1002/hyp.8276.

Guðmundsson, S. *et al.* (2009) 'Similarities and differences in the response to climate warming of two ice caps in Iceland', *Hydrology Research*, 40(5), pp. 495–502. doi: 10.2166/nh.2009.210.

Hock, R. and Jansson, P. (2005) 'Modeling Glacier Hydrology', in Anderson, M. G. and McDonnell, J. J. (eds) *Encyclopedia of Hydrological Sciences 4*. Chichester, UK: John Wiley & Sons, Ltd, pp. 2647–2655.

Jackson, C. R. *et al.* (2016) 'Reconstruction of multi-decadal groundwater level time-series using a lumped conceptual model', *Hydrological Processes*, 30(18). doi: 10.1002/hyp.10850.

Jansson, P., Hock, R. and Schneider, T. (2003) 'The concept of glacier storage: a review', *Journal of Hydrology*, 282(1–4), pp. 116–129. doi: 10.1016/S0022-1694(03)00258-0.

Jobst, A. M. *et al.* (2018) 'Intercomparison of different uncertainty sources in hydrological climate change projections for an alpine catchment (upper Clutha River, New Zealand)', *Hydrology and Earth System Sciences*, 22(6), pp. 3125–3142. doi: 10.5194/hess-22-3125-2018.

Liu, Z. *et al.* (2013) 'Impacts of climate change on hydrological processes in the headwater catchment of the Tarim River basin, China', *Hydrology Research*, 44(5), pp. 834–849.

Mackay, J. D. *et al.* (2015) 'Seasonal forecasting of groundwater levels in principal aquifers of the United Kingdom', *Journal of Hydrology*, 530. doi: 10.1016/j.jhydrol.2015.10.018.

Mackay, J. D. *et al.* (2018) 'Glacio-hydrological melt and run-off modelling: application of a limits of acceptability framework for model comparison and selection', *The Cryosphere*, 12, pp. 2175–2210. doi: 10.5194/tc-12-2175-2018.

Mackay, J. D., Jackson, C. R. and Wang, L. (2014) 'A lumped conceptual model to simulate groundwater level time-series', *Environmental Modelling and Software*, 61, pp. 229–245. doi: 10.1016/j.envsoft.2014.06.003.

Mansour, M. M. *et al.* (2018) 'Estimation of spatially distributed groundwater potential recharge for the United Kingdom', *Quarterly Journal of Engineering Geology and Hydrogeology*. Available at: http://qjegh.lyellcollection.org/content/early/2018/02/08/qjegh2017-051.abstract.

McMillan, H. K. and Westerberg, I. K. (2015) 'Rating curve estimation under epistemic uncertainty', *Hydrological Processes*, 29(7), pp. 1873–1882. doi: 10.1002/hyp.10419.

Nawri, N. *et al.* (2017) *The ICRA atmospheric reanalysis project for Iceland*. Reykjavík, Iceland.

Prudhomme, C. *et al.* (2013) 'Future Flows Hydrology: An ensemble of daily river flow and monthly groundwater levels for use for climate change impact assessment across Great Britain', *Earth System Science Data*, 5(1). doi: 10.5194/essd-5-101-2013.

Sorensen, J. P. R. *et al.* (2014) 'Comparison of varied complexity models simulating recharge at the field scale', *Hydrological Processes*, 28(4), pp. 2091–2102. doi: 10.1002/hyp.9752.

Vetter, T. *et al.* (2015) 'Multi-model climate impact assessment and intercomparison for three large-scale river basins on three continents', *Earth System Dynamics*, 6(1), pp. 17–43. doi: 10.5194/esd-6-17-2015.

Wijngaard, R. R. *et al.* (2017) 'Future changes in hydro-climatic extremes in the Upper Indus, Ganges, and Brahmaputra River basins', *PLOS ONE*. Edited by J. A. Añel, 12(12), p. e0190224. doi: 10.1371/journal.pone.0190224.

---

## Author Comment (AC2) · 20 Dec 2018

[revised manuscript text omitted]

---

## Author Response (AR2)

We would like to thank the handling editor, Markus Hrachowitz and both referees for their helpful review comments while writing this manuscript. We look forward to seeing the final published paper online in the coming weeks.

With best regards

Jonathan Mackay and co-authors